# TOPBENCH: A Benchmark for Implicit Predictive Reasoning in Tabular Question Answering

An-Yang Ji [1 2]   Jun-Peng Jiang [1 2]   De-Chuan Zhan [1 2]   Han-Jia Ye [1 2]

## Abstract

Large Language Models (LLMs) have advanced Table Question Answering, where most queries can be answered by extracting information or simple aggregation. However, a common class of real-world queries is implicitly predictive, requiring the inference of unobserved answers from historical patterns rather than mere retrieval. These queries introduce two challenges: recognizing latent intent and reliable predictive reasoning over massive tables. To assess LLMs in such Tabular questiOn answering with implicit Prediction tasks, we introduce TOPBENCH, a benchmark consisting of 779 samples across four sub-tasks, ranging from single-point prediction to decision making, treatment effect analysis, and complex filtering, requiring models to generate outputs spanning reasoning text and structured tables. We evaluate diverse models under both text-based and agentic workflows. Experiments reveal that current models often struggle with intent recognition, defaulting to just lookups. Deeper analysis identifies that accurate intent disambiguation serves as the prerequisite for leading these predictive behaviors. Furthermore, elevating the upper bound of prediction precision requires the integration of more sophisticated modeling or reasoning capabilities. Our benchmark is available at https://github.com/LAMDA-Tabular/TopBench.

## 1. Introduction

Tabular data represents one of the most prevalent formats in real-world applications, serving as a carrier of rich informa-

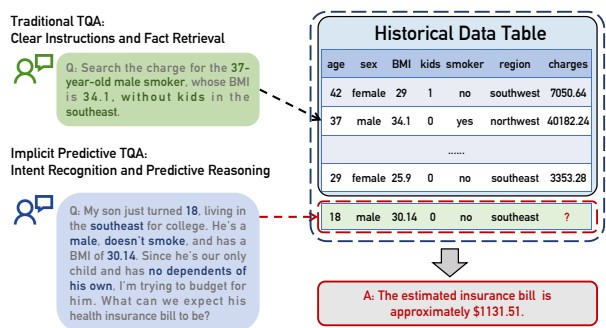

*Figure 1.* **Comparison between Traditional TQA and Implicit Predictive TQA.** Rather than retrieving or aggregating explicit facts based on clear instructions, implicit predictive TableQA requires the model to infer unobserved values.

tion (Cafarella et al., 2008) across critical domains, ranging from financial analysis (Chen et al., 2021b; Zhu et al., 2021; Chen et al., 2022) and medical interpretation (Lee et al., 2022; Shi et al., 2024) to daily record management (Dong et al., 2024). Driven by the growing demand to automatically analyze these structures and extract insights, Large Language Models (LLMs) are increasingly deployed to interpret tabular content, lowering the technical barrier for users to interact with complex structured data (Dibia, 2023; Wang et al., 2024; Jiang et al., 2023).

Tabular Question Answering (TQA) serves as a representative task in this field (Chen et al., 2021a; Yin et al., 2020; Jin et al., 2022). Distinct from unstructured text QA (Rajpurkar et al., 2016; Fisch et al., 2019), TQA requires models to derive answers from the structured schema of tables (Pasupat & Liang, 2015). For instance, for a medical insurance table, a query to search the charge for a specific profile involves information retrieval. More complex questions, such as get the average cost for non-smokers in the southwest, require aggregation to count or summarize the results. However, both tasks remain confined to analyzing explicit facts strictly present within the table.

Importantly, real-world queries with tables extend beyond the simple retrieval of records to the forecasting of unknown outcomes based on historical data. As illustrated in Figure 1, when a user asks about an expected bill for a new profile,

[1]School of Artificial Intelligence, Nanjing University, China [2]National Key Laboratory for Novel Software Technology, Nanjing University, China. Correspondence to: Han-Jia Ye <yehj@lamda.nju.edu.cn>.

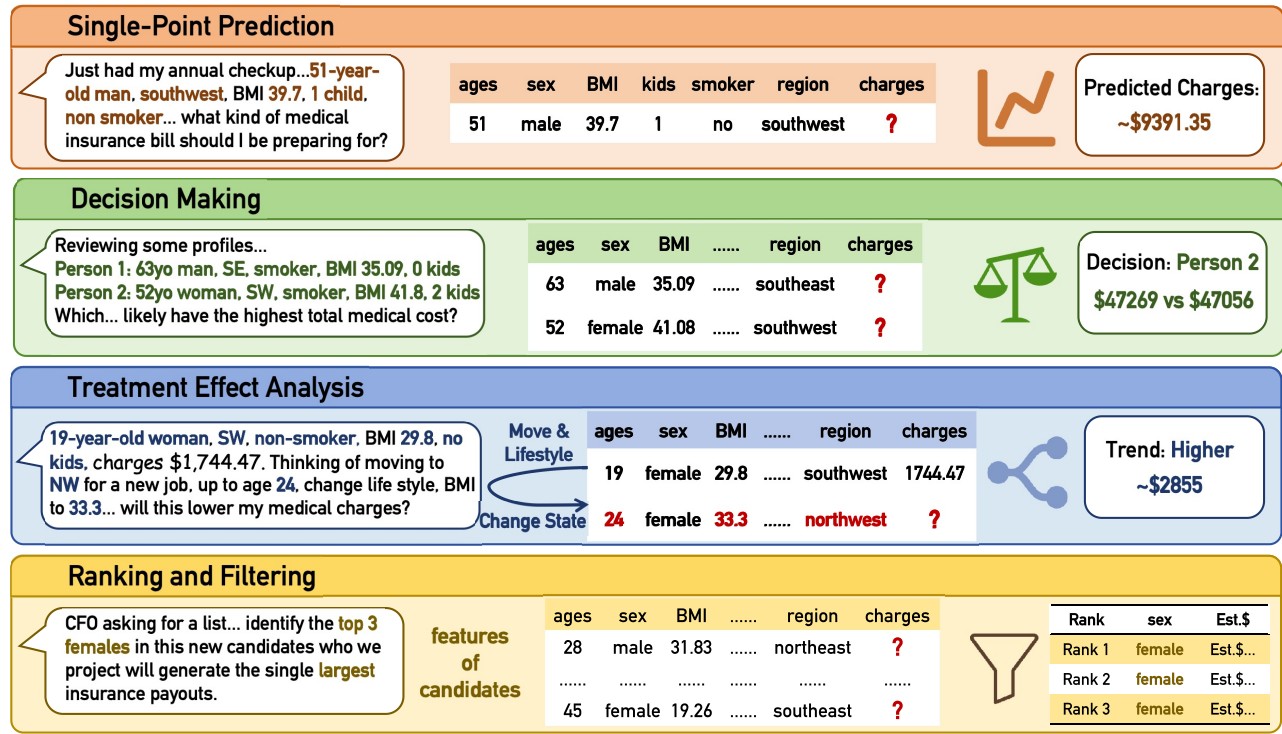

Figure 2. **Overview of TOPBENCH.** TOPBENCH requires inferring unobserved outcomes from historical data across four tasks: Single-Point Prediction, Decision Making, Treatment Effect Analysis, and Ranking and Filtering.

the answer is not present in the database. To address this, the model must first recognize the latent predictive intent by abstracting the textual description into a structured feature set. Subsequently, it needs to perform predictive reasoning over massive historical rows to infer the missing value. This process imposes a dual challenge: the model must abstract the user query into a tabular problem while correctly identifying its predictive nature, and execute precise predictive modeling based on the provided data context.

Crucially, existing benchmarks fail to address this complexity. They mainly focus on retrieving facts from small tables (Pasupat & Liang, 2015; Chen et al., 2020) or following clear execution instructions (Zhong et al., 2018; Yu et al., 2018). On one hand, there is a shortage of datasets that simulate these real-world implicit predictive scenarios. On the other hand, current works fail to analyze the intermediate reasoning process. They are unable to determine whether model failures result from misinterpreting the task intent or from deficiencies in the predictive modeling process itself.

To bridge this gap, we introduce TOPBENCH, a benchmark designed to comprehensively evaluate LLMs on Tabular questiOn answering with implicit Prediction goals. TOP-BENCH comprises 779 high-quality samples spanning three critical domains: healthcare, finance, and daily consulting. We organize the benchmark into four tasks representing

common real-world predictive scenarios: (1) Single-Point Prediction estimates unknown outcomes for specific profiles; (2) Decision Making compares trade-offs to select optimal solutions; (3) Treatment Effect Analysis assesses how interventions alter future results; (4) Ranking and Filtering screens candidates for high-priority targets based on historical data. Models are required to generate both natural language reasoning and structured outputs. We evaluate performance under two paradigms: direct text-based reasoning and agentic workflows utilizing iterative ReAct loops (Yao et al., 2023). To ensure rigorous assessment, we employ a hybrid pipeline combining statistical metrics with an LLM-as-a-Judge approach (Zhang et al., 2024), incorporating evaluation designs specifically adapted to each task and strict verification steps to eliminate extraction hallucinations.

Our experiments across diverse model types reveal that current LLMs face a strict barrier in bridging the gap between intent recognition and predictive modeling. Text-based approaches are frequently constrained by context limits, often degrading from predictive reasoning into hallucinated retrieval. Similarly, agentic workflows often fail to distinguish implicit predictive intents from historical lookups, resulting in low recall for filtering tasks and frequent execution failures. Through further analysis, we identify two pivotal requirements for improvement. First, we validate that effective intent disambiguation serves as the foundational step

to activate predictive modes and prevent models from defaulting to retrieval. Second, elevating performance beyond this baseline requires more than simple tool invocation. It needs superior feature engineering and advanced predictive reasoning mechanisms to accurately model complex data dependencies. While the ultimate goal of TableQA involves handling unstructured requests where data must be retrieved or structured from scratch, TOPBENCH targets a fundamental intermediate challenge. By pairing intent-rich queries with specific historical tables, we isolate the model's capacity to identify latent predictive goals and execute rigorous modeling, laying the necessary groundwork for future autonomous data intelligence.

In summary, our main contributions are as follows:

- We formally define and quantify the task of implicit predictive TQA. This formulation aligns with the complexities of real-world industrial applications, addressing a critical capability gap in existing benchmarks.
- We introduce TOPBENCH, a benchmark that broadens tabular intelligence research toward implicit predictive reasoning. It contains a four-tier task supported by a rigorous automated evaluation pipeline.
- We conduct a comprehensive evaluation on mainstream LLMs. Through systematic failure analysis, we identify key limitations in implicit modeling and execution robustness, offering empirical guidance for future research.

## 2. Related Work

Tabular intelligence spans representation learning, supervised prediction, and question answering. Recent surveys organize tabular representation learning into specialized, transferable, and general-purpose models, clarifying how deep tabular and tabular foundation models differ in their assumptions and deployment scope (Jiang et al., 2026).

### 2.1. Methodologies for Table Question Answering

Methods for interpreting tabular data have evolved significantly. Traditional approaches primarily focused on modifying model architectures with specialized embeddings and attention mechanisms to capture structural dependencies (Herzig et al., 2020; Yin et al., 2020). The advent of LLMs shifted the paradigm toward prompt engineering and in-context learning, where well-organized representations improve interpretation capabilities (Singha et al., 2023; Zhao et al., 2023). To enhance reasoning depth, recent studies leverage chain-of-thought techniques to elicit step-by-step deduction processes (Wei et al., 2022; Tai et al., 2024). Furthermore, integrating external tools such as SQL executors and Python interpreters has become a standard approach to handle complex logical operations and robust data analysis (Cheng et al., 2023; Wang et al., 2025; Chai

et al., 2024). Complementing these inference-time strategies, instruction tuning on diverse tabular datasets further refines model capabilities in structure recognition and fact verification (Zhuang et al., 2024; Zhang et al., 2025).

### 2.2. Evolution of TQA Benchmarks

Existing benchmarks primarily evaluate the ability to retrieve or aggregate explicit information. Foundational datasets like WTQ (Pasupat & Liang, 2015) and SQA (Iyyer et al., 2017) established standards for discrete operations on HTML tables, while TabFact (Chen et al., 2020) introduced verification tasks to assess consistency between text and data. To address the rigidity of extractive answers, generative tasks such as ToTTo (Parikh et al., 2020) and FeTaQA (Nan et al., 2022) required models to synthesize free-form natural language summaries. With the rise of coding capabilities, the field shifted toward evaluating text-to-SQL and code generation performance through benchmarks like WikiSQL (Zhong et al., 2018), Spider (Yu et al., 2018; 2019), and BIRD (Li et al., 2023). Recent agent-centric frameworks such as TableBench (Wu et al., 2025) further simulate complex data analysis workflows. Multimodal tabular benchmarks study complementary problems, including compositional condition QA over table images, tabular-to-visual knowledge transfer, and visual tabular reasoning with privileged structured information (Jiang et al., 2025b; 2024; Lu et al., 2025; Jiang et al., 2025a; Cheng et al., 2025; Jia et al., 2026; Wang et al., 2016; Chu et al., 2024; Wang et al., 2026; Yu et al., 2025). However, a limitation persists across these datasets as they assess the capability to query or summarize existing historical records with clear instructions, but do not evaluate the ability to abstract implicit unstructured intents into tabular tasks or execute rigorous predictive modeling to infer unobserved outcomes, which are key requirements in real-world scenarios.

### 2.3. Tabular Prediction Benchmarks

Tabular prediction has a long independent line of work, from tree-based methods such as XGBoost (Chen & Guestrin, 2016) and CatBoost (Prokhorenkova et al., 2018) to modern deep baselines and tabular foundation models. Recent toolboxes and benchmark analyses emphasize that no single model family dominates across all real datasets, with tree ensembles, nearest-neighbor-inspired deep methods, and foundation models showing data-dependent strengths (Liu et al., 2025a; Ye et al., 2024; 2025b; Yuan et al., 2025). TabPFN (Hollmann et al., 2025) and its scalable variants further demonstrate the promise and limits of in-context tabular prediction, especially under high-dimensional, many-category, or large-scale settings (Liu & Ye, 2025; Ye et al., 2025a). Benchmark suites such as OpenML (Vanschoren et al., 2014), OpenML-CTR23 (Fischer et al., 2023), and TabArena (Erickson et al., 2025) provide standardized com-

parisons for supervised tabular learning, where the target column, feature columns, training data, and evaluation split are specified before modeling begins. In contrast, TOP-BENCH evaluates an intent-grounded predictive TQA setting where the system must first infer the predictive task from a natural-language question over a paired historical table.

## 3. TOPBENCH Dataset

In this section, we present TOPBENCH, a benchmark designed to evaluate implicit predictive reasoning over tabular data. We first introduce the hierarchical task taxonomy that maps real-world queries into formalized predictive problems. Subsequently, we detail the data sourcing and automated synthesis pipeline used to construct high-quality, intent-rich samples. Finally, we provide a statistical analysis of the dataset across domains and task types.

### 3.1. Hierarchical Task Taxonomy

We structure TOPBENCH into four sub-tasks as depicted in Figure 2, representing the most common forms of implicit prediction in real-world scenarios. We formalize implicit predictive TQA as a two-stage inference problem. Given a historical table $T_{\text{hist}} = \{(x_i, y_i)\}_{i=1}^{N}$ containing feature-target pairs and a user query $Q$, the model must first perform intent abstraction to extract a target feature profile $x'$ that does not exist in $T_{\text{hist}}$. Subsequently, the model must execute predictive inference by learning a mapping function $f : x \rightarrow y$ from $T_{\text{hist}}$ to estimate the unknown target $y' = f(x')$. We instantiate this paradigm into the following four diverse tasks derived from real-world user logs.

**Single-Point Prediction.** This serves as the fundamental unit of predictive TQA where the objective is to estimate a single unknown value $y'$ given a specific query profile $x'$. In Figure 2, a user describes a "51-year-old male, southwest, BMI 39.7" and asks for his expected insurance bill. Here, the model must extract these features to form $x'$ and utilize the historical data to predict the specific numerical charge, requiring a direct regression or classification inference.

**Decision Making.** This task extends prediction to comparative reasoning, simulating trade-off analysis. The model receives multiple profiles $x'_1, x'_2, \ldots$ and must determine the optimal choice based on their predicted outcomes. The query in Figure 2 asks the model to compare a "63-year-old male" against a "52-year-old female" to identify who will likely incur the highest medical cost. The model must implicitly predict the charges $y'_1$ and $y'_2$ for both individuals and perform a comparison operation $\text{argmax}(y')$ to derive the final decision.

**Treatment Effect Analysis.** Moving beyond static prediction, this task evaluates causal reasoning by estimating how

*Table 1.* **Statistics of the TOPBENCH Dataset.** The dataset consists of 779 queries derived from 35 unique source tables across three domains. We report the breakdown of queries (#Q.) per task, balanced across predictive objectives (Regression vs. Classification) and interaction perspectives (User vs. Data Holder).

| Task | #Tab. | #Q. | Objective | | Perspective | |
|---|---|---|---|---|---|---|
| | | | Reg. | Class. | User | Holder |
| Single-Point | 35 | 274 | 116 | 158 | 138 | 136 |
| Decision Making | 31 | 186 | 84 | 102 | 93 | 93 |
| Treatment Effect | 12 | 105 | 80 | 25 | 59 | 46 |
| Ranking | 27 | 214 | 104 | 110 | - | 214 |
| Total | 35 | 779 | 384 | 395 | 290 | 489 |

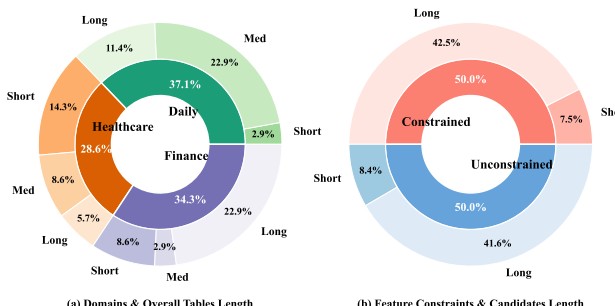

*Figure 3.* **Dataset Distributions.** (a) Domain distribution (Inner Ring) and the corresponding historical table lengths (Outer Ring) defined as Short ($< 1k$), Medium ($1k$-$10k$), and Long ($> 10k$). (b) Distribution of Ranking tasks categorized by filtering constraints (Inner Ring) and the length of candidate lists to be processed (Outer Ring) as Short ($< 100$) and Long ($> 100$).

a change in state variables $\Delta x$ impacts the outcome $\Delta y$. As illustrated in the provided case, a user asks if "moving to the NW" and changing their BMI to "33.3" will lower their charges. The model must effectively perform counterfactual reasoning by predicting the outcome $f(x'_{\text{new}})$ for the modified state and comparing it against the baseline $f(x'_{\text{old}})$ to quantify the treatment effect.

**Ranking and Filtering.** This task addresses the industrial demand for screening massive candidate lists. The model must identify a subset of items $S \subset \{x'_1, \ldots, x'_M\}$ that satisfy explicit constraints while maximizing an implicit predictive criterion. For example, the query requires identifying the "top 3 females" who are projected to generate the "largest payouts". This challenges the model to simultaneously apply feature filtering based on gender and perform batch regression to rank candidates by their predicted costs.

### 3.2. Data Sourcing and Construction

To ensure high-quality predictive reasoning, we curated tables containing genuine correlations from Kaggle across healthcare, finance, and daily consulting domains. These tables vary in scale, ranging from small datasets to industrial-

scale logs, designed to challenge model adaptability. Building on this foundation, we implemented a multi-stage synthesis pipeline to generate natural, intent-rich queries. As shown in Figure 6 of the Appendix, rather than relying on random selection, we employed a logic-driven sampling strategy to select challenging examples with similar feature values or high noise, testing discriminative precision. Furthermore, we adopted a dual-perspective prompting approach that simulates both non-technical "User" descriptions and historical-data-aware "Data Holder" narratives to prevent mechanical outputs. Finally, all samples underwent a hybrid validation process combining independent LLM auditors with expert human review to guarantee the consistency and solvability of the implicit predictive intent.

### 3.3. Dataset Statistics

TOPBENCH comprises 779 high-quality samples derived from 35 unique historical data tables. Each sample pairs a natural language query with a raw CSV file and a structured ground truth JSON. As illustrated in Figure 3(a), the dataset spans three distinct domains: Daily Consulting, Finance, and Healthcare. To test model robustness against data scale, the historical source tables vary significantly in size, ranging from compact datasets with fewer than 1,000 rows to massive industrial logs exceeding 6 million entries.

Beyond domain and scale diversity, TOPBENCH has a structural balance. As shown in Table 1, the predictive objectives are nearly evenly split between regression (384 queries) and classification (395 queries). Furthermore, the benchmark incorporates dual perspectives, featuring 290 user-centric inquiries and 489 data-holder scenarios, thereby testing model adaptability across different role-playing contexts. Finally, for the Ranking and Filtering task shown in Figure 3(b), we ensure a balanced evaluation by equally splitting queries between those requiring explicit additional feature filtering and those relying solely on implicit prediction.

## 4. TOPBENCH Evaluation Methodology

We design a dual-stream evaluation framework to rigorously assess implicit predictive capabilities based on the output modality. The Predictive Reasoning scenarios, comprising Single-Point Prediction, Decision Making, and Treatment Effect Analysis, require natural language explanations with embedded inferential conclusions. Conversely, the Ranking and Filtering task demands the generation of structured CSV files to demonstrate large-scale data manipulation. Our pipeline therefore differentiates between a logic-aware text evaluator and a deterministic structured file evaluator.

### 4.1. Natural Language Reasoning Evaluation

Implicit prediction requires verifying specific outcome values embedded within complex reasoning chains, unlike traditional Table QA tasks that rely on n-gram overlap metrics such as ROUGE (Lin, 2004). We address this by implementing a comprehensive LLM-as-a-Judge pipeline that jointly assesses predictive accuracy and reasoning quality.

**Structured Prediction Extraction.** We employ a specialized judge LLM to parse free-form responses and extract both final conclusions and intermediate predictive values, such as estimates for competing candidates. This step verifies that the model performed the requisite quantitative analysis rather than relying on semantic plausibility. To accommodate the inherent uncertainty in LLM reasoning, we enforce the parallel extraction of predictive intervals alongside single-point estimates. This ensures that valid conservative bounds are evaluated rather than ignored. Importantly, as shown in Figure 7 of the Appendix, we address the issue of judge hallucination by implementing a rigorous verification protocol. By cross-referencing extracted proofs with the original response via fuzzy matching and Natural Language Inference, we ensure that all extracted values are strictly grounded in the model's actual output.

**Reasoning Quality Assessment.** Distinct from numerical accuracy, the judge evaluates logical coherence on a five-point scale. It audits the reasoning chain for fatal flaws such as self-contradiction, circular justification, or the hallucination of constraints not present in the table. This metric acts as a quality gate, ensuring high scores are reserved for clear, evidence-based derivation rather than random guessing.

**Composite Metric Calculation.** We quantify predictive precision using distinct metrics for continuous and discrete outputs. For regression tasks, measuring simple distance to the ground truth is insufficient given that models often generate valid confidence intervals. We define the regression accuracy score ($Acc_{\text{reg}}$) as a weighted combination of pointwise precision and interval coverage, modulated by a penalty for excessive uncertainty:

$$Acc_{\text{reg}} = (\lambda_1 \cdot S_{\text{point}} + \lambda_2 \cdot S_{\text{iou}}) \times P_{\text{width}} \qquad (1)$$

Here, $\lambda_1$ and $\lambda_2$ represent the weights for point estimation and interval overlap, set to 0.6 and 0.4 respectively. The point score $S_{\text{point}}$ is calculated as $\max(0, 1 - \text{NMAE})$ to ensure comparability across datasets. To assess confidence bounds, we calculate the Intersection over Union ($S_{\text{iou}}$) between the predicted and ground truth intervals. We prevent the gaming of this metric via a Width Penalty $P_{\text{width}}$ that decays exponentially if the predicted range exceeds the natural data variance:

$$P_{\text{width}} = \begin{cases} e^{-\alpha\left(\frac{w_{\text{pred}}}{w_{\text{gt}}} - \kappa\right)} & \text{if } w_{\text{pred}} > \kappa \cdot w_{\text{gt}} \\ 1 & \text{otherwise} \end{cases} \qquad (2)$$

where $w$ represents interval width, $\kappa$ is the tolerance threshold set to 2.0, and $\alpha$ controls the decay rate. For Decision Making and Treatment Analysis tasks, we calculate a distinct Decision Score, defined as a binary exact match indicating whether the model successfully identified the optimal candidate or correctly predicted the directional trend (e.g., "increase" vs. "decrease") compared to the ground truth.

### 4.2. Structured Output Evaluation

The Ranking and Filtering scenario simulates industrial batch processing, where the objective is to screen massive datasets and identify candidates meeting complex implicit criteria. This scenario requires the model to act as a data processor, generating a structured file (CSV) containing the filtered or ranked records. Therefore, our evaluation methodology shifts from natural language parsing to a deterministic file-based assessment, focusing on the precision of the retrieval set and the accuracy of the batch predictions.

**Metrics for List Filtering.** For tasks where the goal is to select a subset of candidates (e.g., "Identify all patients with malignant tumors."), we calculate the F1 Score by comparing the set of rows returned by the model against the ground truth subset. This measures the model's ability to strictly adhere to the implicit filtering logic without hallucinating non-existent records or omitting valid candidates.

**Metrics for Complex Ranking.** For tasks involving regression or prioritized sorting (e.g., "Rank the top 10 most profitable companies."), we assess performance across three dimensions. We use Set Recall to verify if the model successfully retrieved the correct top-$k$ items from the full dataset. To evaluate the quality of the sorting, we compute the Normalized Discounted Cumulative Gain (NDCG) (Järvelin & Kekäläinen, 2002), which rewards models for placing high-value items at the very top of the list. Finally, to assess the precision of the predicted numerical values attached to each record, we compute the batch NMAE over the aligned pairs,

$$\text{NMAE}_{\text{batch}} = \frac{1}{N} \sum_{i=1}^{N} \min \left( \frac{|y_{\text{pred}}^{(i)} - y_{\text{gt}}^{(i)}|}{y_{\max} - y_{\min}}, 1.0 \right) \quad (3)$$

where $N$ is the number of successfully matched rows. This formula caps the error for gross outliers, providing a stable measure of how accurately the model predicted the specific attributes of the filtered candidates.

## 5. Experiments

### 5.1. Evaluation Details

**Selected LLMs.** We assess 9 representative models organized into three categories: General LLMs, Reasoning-Enhanced Models, and Tabular Specialists. The General LLMs encompass both latest proprietary frontier models (GPT-5.2 (OpenAI, 2025), Claude Sonnet 4.5 (Anthropic, 2025), and Gemini 3 Flash (Google DeepMind, 2025)) and standard open-weights instruction-tuned models (DeepSeek-V3.2-Instruct (DeepSeek-AI et al., 2025) and Qwen3-Instruct (Yang et al., 2025)). To analyze the impact of test-time compute (Snell et al., 2025), we evaluate Reasoning-Enhanced Models, including DeepSeek-V3.2-Thinking and Qwen3-Thinking. Note that for the Qwen3 family, we utilize the specific Qwen3-235BA22B-2507. Finally, we include Tabular Specialists such as TableLLM-8B (Llama3.1 based) and TableLLM-13B (CodeLlama based) (Zhang et al., 2025) to benchmark the performance gap between generalist capabilities and domain-specific adaptations.

**Inference Paradigms.** To comprehensively assess both intent recognition and predictive modeling capabilities, we employ two inference paradigms. In Text-Based Reasoning, the model receives the query alongside a serialized table string without external tools. Since large datasets are truncated to fit context limits (Liu et al., 2024), this setup requires the model to perform arithmetic and logical inference using only internal parameters. We keep the table header and randomly retain rows when the serialized table exceeds the context budget; Appendix B.3 compares this policy with head and stratified retention. Conversely, the Agentic Framework employs a ReAct loop (Yao et al., 2023) where the model iteratively executes Python code within a Docker sandbox. The model analyzes execution outputs and error logs to dynamically refine its strategy. We apply this framework to all task types when code execution is applicable, and it is required for Ranking and Filtering because this task involves full-scale structure file generation.

### 5.2. Main Results

Table 2 summarizes the performance of various Large Language Models. To ensure alignment, Logic Scores are normalized to the range $[0, 1]$.

**Are LLMs Effective Predictive Reasoners?** Current LLMs demonstrate significant fragility in implicit predictive tasks, with most scores falling below 0.60—a sharp contrast to their proficiency in fact retrieval. Even the leading model, Gemini 3 Flash, achieves only 0.65 accuracy in the fundamental Single Point scenario. This deficiency is amplified in Decision Making and Treatment Effect Analysis, where performance often approximates random guessing. The corresponding intermediate predictions are even worse. Models frequently hallucinate a final conclusion without generating the supporting quantitative estimates, resulting in zero accuracy scores and reflecting a lack of derivation.

**Does the Agentic Framework Help?** Comparing the frameworks reveals distinct behavioral differences. GPT-5.2 demonstrates the benefit of code generation in Treatment

*Table 2.* **Main Evaluation Results on TOPBENCH.** We report performance across four distinct tasks: Single Point Prediction, Decision Making, Treatment Effect Analysis, and Ranking and Filtering. Metrics include Prediction Accuracy (Acc.), Logic Score (Logic), Decision/Trend Accuracy (Dec./Trend), Recall (Rec.), NDCG, NMAE, and F1 Score. The first part displays results for text-based reasoning, where the Ranking and Filtering task is omitted as it requires structured file generation. The second part presents results for the agentic code-execution framework. The best result in each column is highlighted in **bold**.

| Model | Single Point | | Decision Making | | | Treatment Effect | | | Ranking and Filtering | | | |
| | | | | | | | | | Regression | | | Cls. |
| | Acc. | Logic | Acc. | Logic | Dec. | Acc. | Logic | Trend | Rec. | NDCG | NMAE↓ | F1 |
| **Text-Based Reasoning** | | | | | | | | | | | | |
| *General LLMs* | | | | | | | | | | | | |
| GPT-5.2 | 0.55 | 0.72 | 0.22 | 0.76 | 0.57 | 0.18 | 0.76 | 0.51 | - | - | - | - |
| Gemini 3 Flash | **0.65** | 0.75 | **0.42** | **0.78** | **0.62** | **0.43** | **0.78** | **0.62** | - | - | - | - |
| DeepSeek-V3.2-Instruct | 0.59 | 0.75 | 0.32 | 0.76 | 0.57 | 0.35 | 0.77 | 0.59 | - | - | - | - |
| Qwen3-Instruct | 0.57 | **0.76** | 0.36 | 0.75 | 0.53 | 0.28 | 0.74 | 0.55 | - | - | - | - |
| *Reasoning-Enhanced Models* | | | | | | | | | | | | |
| DeepSeek-V3.2-Thinking | 0.57 | 0.75 | 0.27 | 0.75 | 0.61 | 0.35 | 0.76 | 0.59 | - | - | - | - |
| Qwen3-Thinking | 0.44 | 0.52 | 0.23 | 0.53 | 0.39 | 0.33 | 0.52 | 0.46 | - | - | - | - |
| *Tabular Specialists* | | | | | | | | | | | | |
| TableLLM$_{\text{Llama3.1-8B}}$ | 0.38 | 0.52 | 0.17 | 0.62 | 0.46 | 0.17 | 0.46 | 0.52 | - | - | - | - |
| TableLLM$_{\text{CodeLlama-13B}}$ | 0.27 | 0.28 | 0.14 | 0.23 | 0.39 | 0.10 | 0.19 | 0.35 | - | - | - | - |
| **Agentic Code-Execution Framework** | | | | | | | | | | | | |
| *General LLMs* | | | | | | | | | | | | |
| GPT-5.2 | 0.60 | 0.71 | 0.40 | 0.72 | 0.55 | 0.46 | **0.79** | **0.65** | 0.46 | 0.41 | 0.41 | 0.38 |
| Claude Sonnet 4.5 | 0.64 | 0.73 | **0.50** | 0.75 | 0.56 | **0.52** | 0.76 | 0.63 | 0.52 | 0.50 | 0.31 | 0.55 |
| Gemini 3 Flash | **0.66** | 0.72 | 0.46 | 0.74 | **0.65** | 0.46 | 0.77 | **0.65** | 0.53 | **0.50** | 0.30 | **0.58** |
| DeepSeek-V3.2-Instruct | 0.58 | **0.76** | 0.42 | 0.74 | 0.59 | 0.38 | 0.77 | 0.57 | **0.54** | **0.50** | **0.26** | 0.48 |
| Qwen3-Instruct | 0.43 | 0.50 | 0.25 | 0.56 | 0.45 | 0.30 | 0.67 | 0.53 | 0.31 | 0.28 | 0.61 | 0.28 |
| *Reasoning-Enhanced Models* | | | | | | | | | | | | |
| DeepSeek-V3.2-Thinking | 0.61 | 0.75 | 0.40 | **0.78** | 0.58 | 0.43 | 0.78 | **0.65** | 0.50 | 0.46 | 0.35 | 0.38 |
| Qwen3-Thinking | 0.57 | 0.67 | 0.27 | 0.70 | 0.57 | 0.42 | 0.75 | 0.53 | 0.43 | 0.41 | 0.46 | 0.42 |
| *Tabular Specialists* | | | | | | | | | | | | |
| TableLLM$_{\text{Llama3.1-8B}}$ | 0.10 | 0.04 | 0.02 | 0.02 | 0.02 | 0.12 | 0.06 | 0.07 | 0.00 | 0.00 | 1.00 | 0.00 |
| TableLLM$_{\text{CodeLlama-13B}}$ | 0.12 | 0.03 | 0.01 | 0.02 | 0.02 | 0.17 | 0.05 | 0.17 | 0.00 | 0.00 | 1.00 | 0.00 |

Effect Analysis, raising its Trend Score from 0.51 to 0.65 by offloading complex arithmetic. Conversely, Qwen3-Instruct suffers a significant drop in Single-Point Prediction (0.57 to 0.43) within the agentic mode. This degradation stems from the model attempting to retrieve data via code rather than performing the necessary predictive modeling. This highlights a critical alignment failure where the model conflates implicit predictive intent with simple information retrieval, a phenomenon we examine further in Section 6.

**Reasoning Models vs. General Models.** Reasoning-enhanced models, notably Qwen3-Thinking, exhibit severe instability in text-based settings, often underperforming standard baselines. In over 50% of cases, the model enters repetitive loops that exhaust the context window. This suggests that despite test-time compute scaling, it struggles to main-

tain logical coherence over long tabular inputs, frequently degenerating into circular logic.

**Challenges in High-Throughput Filtering.** For the Ranking and Filtering task, which needs large-scale structured file generation, precise output remains a substantial challenge. While Gemini 3 Flash leads in classification with a F1 score of 0.58, regression precision is generally poor, with average NMAE scores ranging from 0.30 to 0.40. DeepSeek-V3.2-Instruct achieves the lowest error (0.26), demonstrating relative high performance during batch processing.

**Limitations of Tabular Specialists.** Domain-specific models significantly underperform generalist LLMs across all tasks. This shortfall likely stems from their specialized training on explicit information extraction and code generation

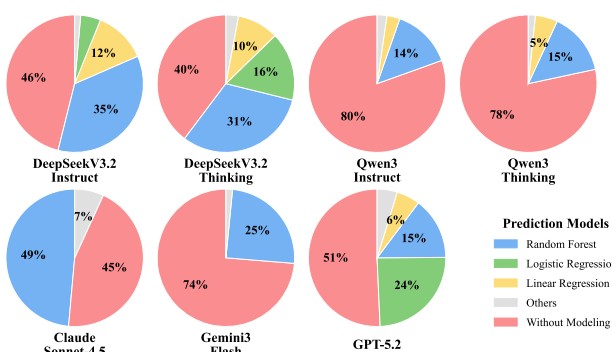

*Figure 4.* **Distribution of Predictive Tool Usage.** The chart illustrates the frequency with which different LLMs invoke machine learning libraries versus simple data manipulation methods. It also highlights the most frequently selected algorithm for each model.

for simple calculation, lacking the broader world knowledge necessary to interpret implicit predictive intents. In filtering regimes, these models exhibit catastrophic failure with near-zero recall due to invalid code execution. This confirms that traditional "Table QA" training objectives are insufficient and that robust code generation is a strict prerequisite for industrial-grade tabular workflows.

## 6. Further Analysis

In this section, we investigate the underlying causes of performance limitations. Specifically, we examine whether current models can effectively translate implicit unstructured queries into structured tabular prediction tasks and how different modeling strategies impact final precision. More specific results can be found in Appendix D.

### 6.1. Analysis of Agentic Predictive Modeling Behaviors

To understand the mechanisms driving performance in the agentic framework, we analyze code generation patterns to determine if models utilize the sandbox for rigorous predictive modeling or merely default to data retrieval.

**Tendency for Predictive Modeling.** Figure 4 illustrates a distinct divergence in tool-use strategy. DeepSeek actively imports libraries such as `scikit-learn` to train predictive models in over 60% of scenarios. In contrast, Qwen3 defaults to `pandas`-based filtering or heuristic arithmetic, often failing to recognize the implicit predictive intent. Regarding algorithm selection, Random Forest (Breiman, 2001) and Logistic Regression (Hosmer et al., 2013) are dominant. This preference likely stems from the distribution of data science code in the pre-training corpus, leading models to favor algorithms that are robust and require minimal hyperparameter tuning.

**Impact of Code Execution on Predictive Precision.** Figure 5 contrasts average performance metrics between in-

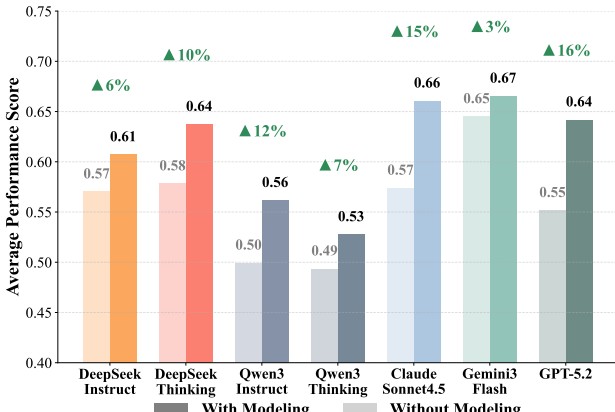

*Figure 5.* **Performance Impact of Predictive Modeling.** We compare average scores across Single Point (Accuracy), Decision (Decision Score), and Treatment Effect (Trend Score) tasks. "With Modeling" means LLMs use the predictive model. "Without Modeling" means they default to data retrieval or simple aggregation.

*Table 3.* **Semantic Information Ablation.** Orig. vs. +Info. Single, Decision, and Treat. use accuracy-style metrics; Rank(Reg.) uses NMAE↓, and Rank(Cls.) uses F1.

| Model | Single | | Decision | | Treat. | | Rank(Reg.) | | Rank(Cls.) | |
|---|---|---|---|---|---|---|---|---|---|---|
| | Orig. | +Info | Orig. | +Info | Orig. | +Info | Orig. | +Info | Orig. | +Info |
| **Text-Based Reasoning** | | | | | | | | | | |
| GPT-5.2 | 0.55 | 0.59 | 0.57 | 0.54 | 0.51 | 0.52 | - | - | - | - |
| Gemini 3 Flash | 0.65 | 0.66 | 0.62 | 0.62 | 0.62 | 0.69 | - | - | - | - |
| DeepSeek-V3.2 | 0.59 | 0.59 | 0.57 | 0.57 | 0.59 | 0.57 | - | - | - | - |
| Qwen3-Instruct | 0.57 | 0.59 | 0.53 | 0.55 | 0.55 | 0.62 | - | - | - | - |
| **Agentic Code-Execution Framework** | | | | | | | | | | |
| GPT-5.2 | 0.60 | 0.64 | 0.55 | 0.53 | 0.65 | 0.68 | 0.41 | 0.35 | 0.38 | 0.48 |
| Gemini 3 Flash | 0.66 | 0.69 | 0.65 | 0.61 | 0.65 | 0.64 | 0.30 | 0.31 | 0.58 | 0.57 |
| DeepSeek-V3.2 | 0.58 | 0.60 | 0.59 | 0.59 | 0.57 | 0.68 | 0.26 | 0.24 | 0.48 | 0.55 |
| Qwen3-Instruct | 0.43 | 0.56 | 0.45 | 0.46 | 0.53 | 0.50 | 0.61 | 0.55 | 0.28 | 0.33 |

stances where models explicitly trained a predictor versus those relying solely on internal reasoning or simple calculation. The data confirms a positive correlation between code-based modeling and task success across all categories. Notably, DeepSeek, which has a higher modeling frequency, significantly outperforms Qwen3. While Gemini 3 Flash achieves a relatively high baseline score of 0.65 through strong intrinsic reasoning, explicit code invocation still yields further performance gains. This indicates that even for advanced models capable of approximate context processing, rigorous predictive modeling remains essential for precision-critical tasks.

### 6.2. Ablation on Intent Recognition and Semantic Understanding

To test whether failures come from missing the predictive intent, we inject semantic hints into the instruction: target column, task type, and feature descriptions. Table 3 compares the original models with these info-enhanced variants.

*Table 4.* **Predict-Only Baseline.** The ensemble receives the gold structured target/profile and is compared with the strongest Gemini agentic E2E setting.

| Method | Single Acc. | Decision Dec. | Treat. Trend |
|---|---|---|---|
| Gemini 3 Flash (agentic E2E) | 0.66 | 0.65 | 0.65 |
| Predict-only ensemble | **0.76** | **0.72** | **0.69** |

**Correction of Intent Misalignment.** The clearest gains in Table 3 occur when the original model likely chooses the wrong task framing. In the agentic setting, Qwen3-Instruct improves on Single Point prediction from 0.43 to 0.56, DeepSeek-V3.2 improves on Treatment Effect Analysis from 0.57 to 0.68, and GPT-5.2 improves on Single Point prediction from 0.60 to 0.64. Text-based gains are smaller but still visible. These results indicate that a major failure mode is not coding ability, but failure to recognize prediction as the required operation.

**Benefits for Complex Batch Processing.** For Ranking and Filtering, evaluated only in the agentic setting, semantic information helps feature selection in noisy candidate pools. GPT-5.2 improves on both metrics (NMAE 0.41→0.35, F1 0.38→0.48), as do DeepSeek-V3.2 (0.26→0.24, 0.48→0.55) and Qwen3-Instruct (0.61→0.55, 0.28→0.33). Gemini 3 Flash changes little, suggesting that stronger models may already infer much of the schema semantics from the raw table.

**Precision Barrier in Fine-Grained Decisions.** The Decision Making columns show a different pattern. Semantic information brings little benefit and sometimes hurts performance. Gemini 3 Flash drops from 0.65 to 0.61 in the agentic setting, and GPT-5.2 drops from 0.55 to 0.53. Candidate pairs are intentionally similar, so recognizing the intent is necessary but not sufficient; the remaining bottleneck is prediction precision, feature engineering, and the tabular model. This aligns with recent tabular prediction studies showing that stronger specialized predictors and TabPFN-style models can improve prediction quality, but remain sensitive to data scale, dimensionality, and preprocessing choices (Liu et al., 2025a; Liu & Ye, 2025; Ye et al., 2025a).

### 6.3. Ablation on Predictive Modeling Capacity

Table 4 examines the prediction upper bound after the target and feature profile are provided. The predict-only ensemble selects among HistGradientBoosting (Pedregosa et al., 2011), XGBoost (Chen & Guestrin, 2016), LightGBM (Ke et al., 2017), CatBoost (Prokhorenkova et al., 2018), Extra-Trees (Geurts et al., 2006), and TabPFN (Hollmann et al., 2025), following the observation that robust tabular performance often requires adaptive model selection over heterogeneous datasets (Liu et al., 2025a; Ye et al., 2024). It is not an end-to-end TOPBENCH system, but a diagnos-

tic reference for the prediction module that current agentic workflows try to construct through code.

**Targeted Data Processing and Model Selection.** Compared with the strongest Gemini agentic end-to-end setting, predict-only improves Single Point from 0.66 to 0.76, Decision Making from 0.65 to 0.72, and Treatment Effect from 0.65 to 0.69. These gains suggest that stronger prediction requires a task-specific tabular pipeline, not merely invoking a generic estimator. The agent must select useful features, encode categorical variables, handle missing values and scale, and choose model families suited to the target type and table size. Small errors in these steps can directly change a numerical estimate or candidate preference.

**Contextual Comparison and Self-Correction.** The predict-only baseline is still far from perfect, showing that TOPBENCH also requires reasoning over difficult prediction contexts. Noisy distributions, long-tail targets, class imbalance, and similar candidates require models to compare scenario-level predictions, check whether effect directions or candidate orderings are plausible, and revise preprocessing or modeling choices when outputs conflict with table evidence. LLM-assisted tabular ensembling is an early step toward such adaptive comparison, but our setting additionally requires recovering the structured task from an implicit user request (Liu et al., 2025b). Thus future agents need stronger predictive capability across the full workflow: targeted data processing, appropriate model selection, contextual comparison, and self-correction.

## 7. Conclusion

In this work, we introduced TOPBENCH, a benchmark for studying implicit predictive reasoning in TQA beyond explicit fact retrieval. Our evaluation exposes a critical capability gap: current models often struggle to distinguish between retrieval and prediction intents, frequently defaulting to simple lookups rather than constructing appropriate predictive procedures. At the same time, our study operates under a specific constraint where the relevant historical table is explicitly paired with each query. This design avoids mixing predictive reasoning errors with failures in data search or table construction, allowing us to more clearly measure whether a model can identify the intended predictive target and use historical evidence properly. This design isolates predictive reasoning as an intermediate step toward realistic data intelligence. In open-ended applications, user requests are often unstructured, requiring systems to not only reason over provided data but also autonomously retrieve relevant historical records or structure undefined information from scratch. By formalizing this setting and providing a rigorous automated evaluation pipeline, TOPBENCH offers a principled testbed for future research on the full pipeline from data discovery to predictive inference.

## Acknowledgements

This work was supported in part by the Key Program of Jiangsu Science Foundation (No. BK20243012), the National Natural Science Foundation of China (NSFC) (No. 62522605), the Fundamental and Interdisciplinary Disciplines Breakthrough Plan of the Ministry of Education of China (No. JYB2025XDXM118), the "111 Center" (No. B26023), and the Collaborative Innovation Center of Novel Software Technology and Industrialization.

## Impact Statement

This paper advances tabular QA with implicit prediction tasks. TOPBENCH is an evaluation benchmark rather than a deployed decision system, but the setting touches high-stakes domains such as healthcare and finance. Overconfident predictive answers may amplify spurious correlations, distribution shift, or biased historical records. We therefore recommend using the benchmark to diagnose model limitations and requiring domain review before any real-world predictive decision is made.

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

## A. Overview of the Appendix

This Appendix provides supporting details for TOPBENCH. It is organized as follows:

- **Appendix B** gives dataset details, including source-table statistics, table-shape analysis, truncation analysis, the data synthesis pipeline, and the human authenticity audit.
- **Appendix C** gives evaluation details, including hallucination verification, judge configuration, robustness checks, metric sensitivity, prompt templates, and task metrics.
- **Appendix D** reports additional experimental results for ranking metrics, schema information, output integrity, and predictive tool usage.
- **Appendix E** provides qualitative case studies and error analysis.

## B. Dataset Details and Datasheet

In this section, we provide granular statistics of the TOPBENCH dataset, a detailed list of source tables, and the specific methodologies used for data synthesis.

### B.1. Expanded Dataset Statistics

While the main paper outlines the distribution across domains and tasks, here we provide a detailed breakdown of the tabular context scale. As shown in Figure 3, TOPBENCH challenges models with highly varying context lengths.

**Historical Context Lengths.** We categorize the 35 source tables into Short ($< 1$k rows), Medium (1k–10k rows), and Long ($> 10$k rows). Table 5 presents the detailed distribution of rows and tokens for each domain.

*Table 5.* **Detailed Statistics of Historical Table Scales.** We report the row count range and the average token count (using the Tiktoken tokenizer) for tables in each domain.

| Domain | Table Length Distribution (# Tables) | | | Scale Metrics | |
|---|---|---|---|---|---|
| | Short ($< 1$k) | Med. (1k-10k) | Long ($> 10$k) | Max Rows | Avg. Tokens |
| Daily Consulting | 1 | 8 | 4 | 239671 | 6368066 |
| Finance | 3 | 1 | 8 | 6778266 | 7755376 |
| Healthcare | 5 | 3 | 2 | 1020922 | 11257450 |
| **Total** | **9** | **12** | **14** | **6778266** | **8118243** |

**Ranking Candidate Scales.** For the Ranking and Filtering task, the difficulty is determined by the size of the candidate list that the model must process in the current context.

- **Short List ($< 100$ candidates):** 34 queries (approx. 16%).

- **Long List (100–200 candidates):** 180 queries (approx. 84%).

This distribution confirms that TOPBENCH predominantly evaluates the capability to handle long-context inputs in agentic workflows.

### B.2. Table Shape and Scale Analysis

To make the scale variation explicit, we also summarize query-level table contexts. Counting the table context attached to each query, TOPBENCH spans 292 table instances. Row counts range from 160 to 6,778,266, with a median of 2,968 and a mean of 85,222.3. Column counts range from 5 to 55, with a median of 13 and a mean of 17.7. Overall, 47.3% of table contexts contain missing values, with an average missing-cell ratio of 2.85%. Categorical features appear in 78.4% of contexts.

*Table 6.* **Query-Level Table Structure.** Column buckets are computed over query-level table contexts.

| Statistic | Min | Median | Mean | Max | Miss. Tbl. | Cat. Tbl. |
|---|---|---|---|---|---|---|
| Rows | 160 | 2,968 | 85,222.3 | 6,778,266 | 47.3% | - |
| Columns | 5 | 13 | 17.7 | 55 | - | 78.4% |

*Table 7.* **Distribution by Table Shape.** Row buckets are Short ($\leq$1k), Medium (1k–100k), and Long (>100k). Column buckets are Narrow ($\leq$10), Mid (11–30), and Wide (>30).

| Shape Axis | Row Scale | | | Column Width | | |
|---|---|---|---|---|---|---|
| | Short | Medium | Long | Narrow | Mid | Wide |
| Share | 32.2% | 54.5% | 13.4% | 33.6% | 52.4% | 14.0% |

### B.3. Truncation Ablation

For text-based reasoning, 136 of 779 queries (17.5%) require truncation. Restricting to the three predictive reasoning tasks, 136 of 565 queries (24.1%) require truncation: 64/274 for Single-Point Prediction, 42/186 for Decision Making, and 30/105 for Treatment Effect Analysis. We compare three row-retention strategies while always preserving the table header: retaining head rows, random retention, and stratified retention across row chunks.

### B.4. Multi-Stage Data Synthesis Pipeline

To ensure that the generated queries reflect real-world predictive intent rather than simple fact retrieval, we designed a rigorous three-stage synthesis pipeline. The workflow, illustrated in Figure 6, transitions from logic-driven sampling to dual-perspective generation and concludes with a hybrid validation mechanism.

**Stage 1: Foundation Curation and Logic-Driven Sampling.** Unlike traditional benchmarks that sample rows randomly, we employ a logic-driven strategy to maximize difficulty and realism. First, we perform semantic feature engineering to identify high-cardinality columns (e.g., specific names or complex IDs), marking them for special handling to prevent token exhaustion. For the **Decision Making** task, we utilize a "Hard Negative Sampling" approach. Instead of randomly pairing options, we compute the similarity of target values between candidates and select pairs with minimal ground-truth differences. This forces the model to rely on precise predictive modeling rather than rough heuristics. Similarly, for **Ranking and Filtering**, we curate high-noise candidate pools where only a small fraction of entries satisfy the implicit predictive criteria, testing the model's ability to filter massive datasets.

**Stage 2: Task Construction.** To avoid the mechanical tone often found in synthetic datasets, we developed a dual-perspective prompting framework.

- **User Persona:** For single-point prediction and decision-making, prompts simulate a non-technical individual facing a real-world dilemma. The generator is strictly instructed to weave feature values into a coherent narrative (e.g., describing a patient's symptoms or a car's condition) without using technical column names or JSON formatting. This ensures the query retains the ambiguity and casualness of natural human speech.
- **Data Holder Persona:** For complex filtering tasks, the persona shifts to a domain expert (e.g., a Hiring Manager or Risk Officer). These queries explicitly reference historical archives and articulate business-centric goals (e.g., "identifying top candidates for high-risk profiles"), requiring the model to bridge the gap between business language and statistical operations.

**Stage 3: Hybrid Validation with Reward Modeling.** The final stage ensures the validity and solvability of the generated queries through a two-step verification process. First, we deploy a specialized **LLM Reward Model** to audit the generated text. This model scores samples based on four criteria: completeness of feature inclusion, numerical accuracy (allowing for minor formatting variations), absence of hallucinations, and naturalness of expression. We enforce a strict quality threshold (Score > 5.0); samples falling below this score trigger a **Fallback & Simplification** mechanism, which regenerates the query with reduced complexity (e.g., fewer comparison options). Finally, samples passing the automated audit undergo expert human review (Human-In-The-Loop) to confirm that the implicit predictive intent is logically solvable given the provided history.

*Table 8.* **Average Performance by Row Scale.** Values are averaged over per-model means.

| Metric | Short | Med. | Long |
|---|---|---|---|
| Single-Point, text-based Acc. | 0.5645 | 0.5009 | 0.5858 |
| Single-Point, with-tool Acc. | 0.5624 | 0.5091 | 0.5162 |
| Decision Making, text-based Dec. | 0.5910 | 0.5019 | 0.5377 |
| Decision Making, with-tool Dec. | 0.4801 | 0.4624 | 0.4927 |
| Treatment Effect, text-based Trend | 0.4703 | 0.6221 | 0.5333 |
| Treatment Effect, with-tool Trend | 0.4497 | 0.5907 | 0.5412 |

*Table 9.* **Average Performance by Column Width.** There is no wide-column Treatment Effect subset in the current benchmark.

| Metric | Narrow | Mid | Wide |
|---|---|---|---|
| Single-Point, text-based Acc. | 0.5255 | 0.5298 | 0.5990 |
| Single-Point, with-tool Acc. | 0.4922 | 0.5440 | 0.5146 |
| Decision Making, text-based Dec. | 0.4769 | 0.5253 | 0.6778 |
| Decision Making, with-tool Dec. | 0.3860 | 0.4766 | 0.6256 |
| Treatment Effect, text-based Trend | 0.5396 | 0.5613 | - |
| Treatment Effect, with-tool Trend | 0.4778 | 0.5884 | - |

## B.5. Human Authenticity Audit

To check whether generated queries preserve realistic predictive intent, five annotators audited 48 sampled queries. Each query was rated on authenticity, naturalness, and plausibility with respect to the paired table and task setup. Table 13 reports the average scores.

## C. Detailed Evaluation Protocols

This section provides the rigorous definitions of the metrics used in TOPBENCH and details the multi-stage verification protocols employed in our LLM-as-a-Judge pipeline.

### C.1. Hallucination Verification Mechanism

To ensure that the values extracted by the Judge LLM are faithful to the model's original free-form response, we implement a strict **Hallucination Verification Pipeline**. As illustrated in Figure 7, this mechanism filters out "hallucinated" extractions where the Judge might infer a value that the model did not explicitly state. The verification process consists of three hierarchical layers designed to balance precision and flexibility:

1. **Text Standardization and Hybrid Surface Matching:** Before comparison, raw responses undergo aggressive normalization: stripping LaTeX formatting (e.g., converting $\text{1.5k}$ to $1.5k$), unifying Unicode symbols, and mapping natural language numerals (e.g., "three") to digits via a lookup table. The system then attempts to verify the extracted proof quote using a dual-strategy approach:
   - *Spliced Matching*: Handles citations with ellipses (e.g., "The value... is high") by verifying the sequential existence of split segments.
   - *Fuzzy Token Alignment*: Uses a token subsequence algorithm (threshold $\geq 0.8$) to tolerate minor morphological variations such as tense changes ("increase" vs. "increased") or pluralization, provided the core semantic tokens align.

2. **Numeric Parsing and Interval Logic:** For regression tasks, we deploy a specialized `NumberParser` to verify mathematical equivalence between the extracted structure and the text. This module:
   - *Unit Conversion*: Maps domain-specific suffixes (e.g., "1.5k" $\rightarrow$ 1500, "20%" $\rightarrow$ 0.2, "5bn" $\rightarrow$ $5 \times 10^9$).
   - *Interval Derivation*: Validates implied intervals. For instance, if the text states "$10 \pm 2$", the system dynamically computes the range $[8, 12]$ to verify against the extracted bounds.
   - *Boundary Checks*: Enforces strict distinction between lower and upper bounds (e.g., "more than 100" confirms 100 as a lower bound but rejects it as an upper bound).

3. **Semantic Entailment (NLI Fallback):** If structural and numeric matching fail—often due to high-level summarization—we activate a secondary LLM agent to perform Natural Language Inference (NLI). This agent determines if the

*Table 10.* **Model-by-Row-Scale Cross-Tabulation for Text-Based Reasoning.** Entries are Short/Medium/Long scores.

| Model | Single | Decision | Treatment |
|---|---|---|---|
| DeepSeek-V3.2-Ins | 0.63/0.53/0.65 | 0.57/0.58/0.55 | 0.47/0.70/0.57 |
| DeepSeek-V3.2-Think | 0.61/0.50/0.68 | 0.67/0.59/0.57 | 0.50/0.72/0.50 |
| Gemini3.0-Flash | 0.73/0.59/0.71 | 0.74/0.57/0.60 | 0.56/0.70/0.57 |
| GPT-5.2 | 0.54/0.54/0.58 | 0.63/0.52/0.60 | 0.41/0.56/0.57 |
| Qwen3-Ins | 0.62/0.54/0.57 | 0.59/0.51/0.50 | 0.47/0.67/0.47 |
| Qwen3-Think | 0.52/0.40/0.43 | 0.43/0.37/0.41 | 0.47/0.47/0.43 |
| TableLLM-8B | 0.36/0.36/0.43 | 0.57/0.38/0.48 | 0.47/0.61/0.47 |
| TableLLM-13B | 0.22/0.27/0.36 | 0.39/0.37/0.43 | 0.34/0.30/0.43 |

*Table 11.* **Truncation-Affected Subset.** Entries are Head/Random/Stratified. Scores are reported only on queries whose serialized table exceeds the context budget.

| Model | Single | Decision | Treatment |
|---|---|---|---|
| DeepSeek-V3.2-Ins | 0.56/0.63/0.65 | 0.55/0.57/0.57 | 0.60/0.60/0.53 |
| DeepSeek-V3.2-Think | 0.59/0.57/0.56 | 0.60/0.48/0.62 | 0.40/0.67/0.67 |
| Qwen3-Ins | 0.55/0.58/0.58 | 0.55/0.52/0.48 | 0.50/0.53/0.53 |
| GPT-5.2 | 0.58/0.53/0.59 | 0.64/0.60/0.60 | 0.50/0.50/0.60 |

extracted claim is *logically entailed* by the source text. This step allows for valid semantic paraphrasing (e.g., equating "skyrocketed" with "increased significantly") while rigorously rejecting unsupported inferences or hallucinated data points not present in the original response.

**Structured Extraction Schema.** To standardize evaluation across diverse tasks, the Judge extracts predictions into specific JSON schemas. Figure 8 illustrates the extraction formats for Single-Point Prediction, Decision Making, and Treatment Effect Analysis. Note that for complex reasoning tasks, we extract both the final conclusion (e.g., "trend") and the intermediate scenario-specific predictions to verify the chain of thought.

### C.2. Judge Configuration, Robustness, and Prompts

The main judge used for structured extraction and logic scoring is DeepSeek-V3.2-Instruct. In the public release, this is configured through the DeepSeek chat endpoint and can be overridden by DEEPSEEK_JUDGE_MODEL_ID. We use GPT-5.2 as an alternate judge for robustness checks. Prompt templates and decoding settings are fixed by task and mode. For Ranking and Filtering, evaluation is deterministic and file-based, so no judge model is used.

The following are the core prompt templates exposed in the release under src/topbench/prompts. We include the text-based inference prompts, the agentic workflow prompts, and the main extraction prompt; task-specific judge schemas are shown in Figure 8.

**Text-Based Reasoning Prompt**

```
# System prompt

Here is the preview/content of the history data:

$history_table

# User prompt, single-CSV tasks

$query

# User prompt, dual-CSV tasks

$query

$current_table
```

*Table 12.* **Full Text-Based Benchmark Under Different Truncation Policies.** Entries are Head/Random/Stratified. Effects are bounded because most queries do not require truncation.

| Model | Single | Decision | Treatment |
|---|---|---|---|
| DeepSeek-V3.2-Ins | 0.57/0.58/0.59 | 0.56/0.57/0.57 | 0.59/0.59/0.57 |
| DeepSeek-V3.2-Think | 0.56/0.53/0.54 | 0.56/0.51/0.56 | 0.54/0.62/0.62 |
| Qwen3-Ins | 0.56/0.57/0.56 | 0.54/0.53/0.52 | 0.54/0.55/0.55 |
| GPT-5.2 | 0.53/0.53/0.55 | 0.56/0.55/0.56 | 0.50/0.50/0.57 |

*Table 13.* **Human Authenticity Audit.** Scores are averaged over a five-annotator audit of 48 sampled queries.

| Dimension | Average Score |
|---|---|
| Authenticity | 4.00 / 5 |
| Naturalness | 4.00 / 5 |
| Plausibility | 4.75 / 5 |

---

**Agentic Workflow Prompt (Single CSV)**

```
The history data file is located at: history.csv
(Note: The file is mounted in your environment, use 'history.csv' directly)
Here are the columns of the file:
[$column_preview]

IMPORTANT: You MUST use the 'CodeRunner' tool to read the file to inspect the data content.
You need to give the answer within $max_iterations rounds.
```

---

**Agentic Workflow Prompt (Dual CSV)**

```
You have access to two csv files in your environment:
1. 'history.csv'
   - Columns: $history_columns
2. 'current.csv'
   - Columns: $current_columns

Note: These files are mounted, use their filenames directly.
The data provided above are ONLY column names. DO NOT hallucinate data rows.
You MUST use the CodeRunner tool to read the files (e.g., pd.read_csv) to inspect the actual data content.

CRITICAL REQUIREMENT:
1. You MUST process the data and save the final results into a file named 'result.csv'.
2. The 'result.csv' MUST contain the exact columns matching history.csv format.
3. Do not just print the result, you must save it to 'result.csv' using pandas to_csv().$prompt_extras
You need to give the answer within $max_iterations rounds.
```

---

**Judge Regression Extraction Prompt**

```
You are an expert evaluator for a tabular data prediction task (REGRESSION).

Input Data:
[Query]:
{query}

[Model Response]:
{response}

[Ground Truth]:
{gt_str}

[Dataset Metadata]:
{dataset_meta_str}

---
### YOUR TASKS
```

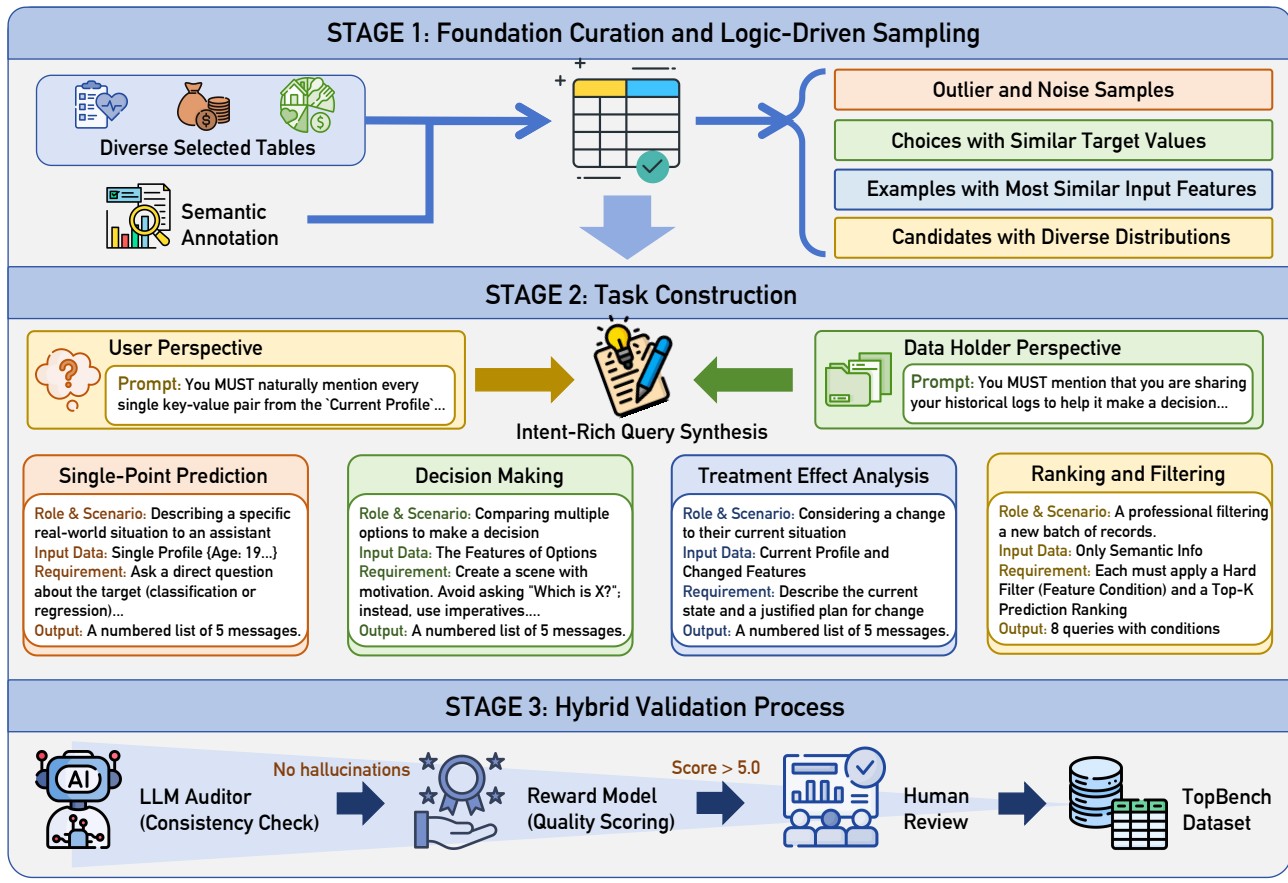

*Figure 6.* **The Multi-Stage Data Synthesis Pipeline.** The process begins with *Foundation Curation*, where logic-driven sampling selects challenging data points (e.g., hard negatives with similar feature values). In *Task Construction*, we employ a dual-perspective prompting strategy—simulating both non-technical users and professional data holders—to generate intent-rich queries across four sub-tasks. Finally, the *Hybrid Validation* phase filters samples using an LLM-based reward model followed by expert human verification to ensure solvability and alignment.

```
1. Prediction Extraction (STRICT):
   - Extract the final numerical prediction or interval.
   - CRITICAL RULE FOR VAGUE NUMBERS:
     - If the text says "2 million+", "over 500k", or "approx 10%",
       you must extract the visible number (e.g., 2000000, 500000, 10).
     - DO NOT make up a precise number to represent the "+".
     - Unless the precise number is explicitly stated in another part
       of the text.
   - CRITICAL RULE FOR INTERVALS: Do NOT narrow down or calculate.
     Extract the EXACT boundaries mentioned.
   - If the prediction value does not exist, set as null.
   {SYMBOL_RULE}

2. Proof Extraction (CRITICAL):
   - You MUST copy the proof_quote VERBATIM from the [Model Response].
   - The numbers in predicted_interval MUST be visibly identical to
     the numbers in this quote.
   - For intervals, your quote MUST contain the text for BOTH the lower
     and upper bounds.
   - DO NOT add property names, keys, or prefixes.
   {ANTI_HALLUCINATION_EXAMPLES}

{logic_block}

---
### OUTPUT JSON FORMAT
{
```

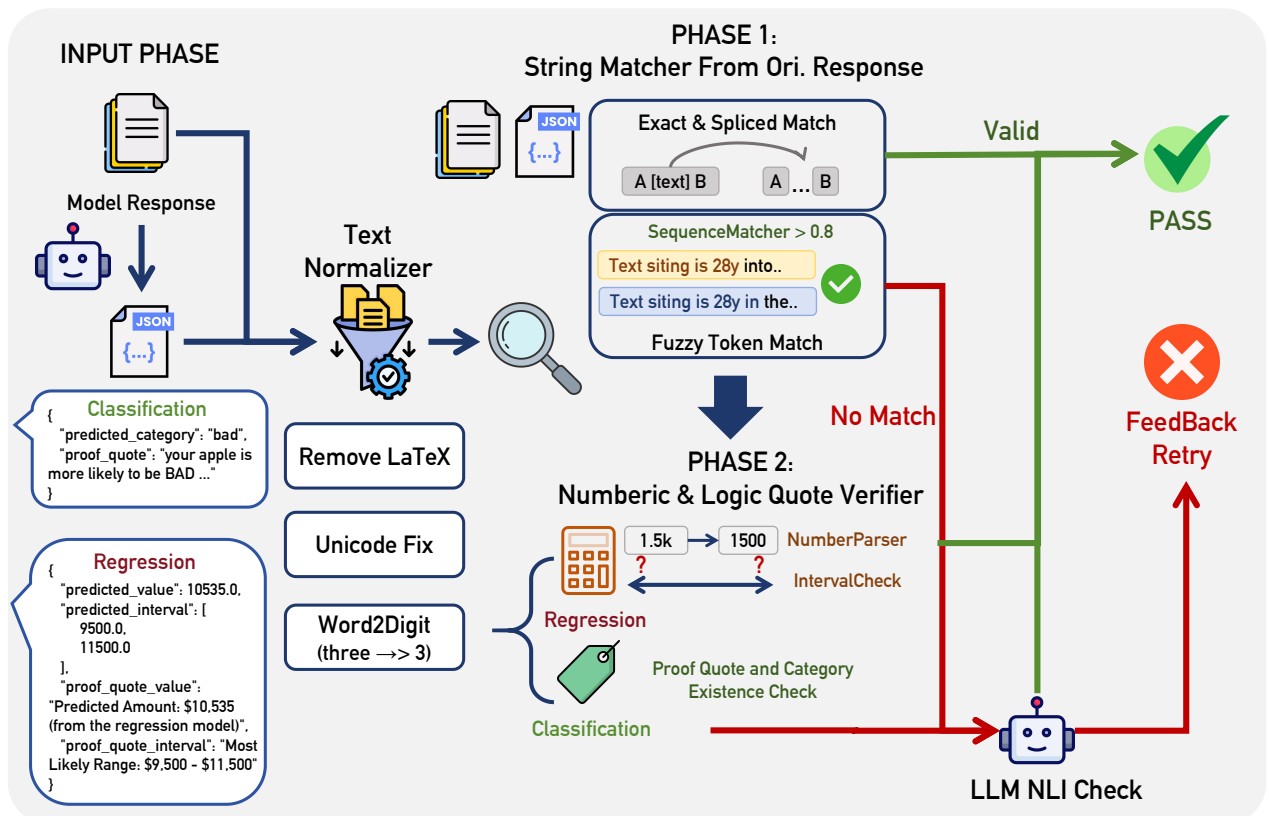

*Figure 7.* **The Hallucination Verification Pipeline.** To ensure data integrity, the extraction process employs a cascade of verification modules. *Phase 1* performs aggressive text normalization followed by surface-level string matching. If direct matching fails, *Phase 2* activates deep logic verifiers, including a numerical parser for unit conversion and an NLI agent to confirm semantic entailment.

```
"prediction_payload": {
  "predicted_value": number or null,
  "predicted_interval": [min, max] or null,
  "proof_quote_value": "exact substring from response containing the value. Don't add any other words",
  "proof_quote_interval": "exact substring from response containing the interval. Don't add any other words"
},
"logic_assessment": {
  "logic_score_raw": 0-5,
  "detected_flaws": ["List strings from STEP 1 (e.g. 'Self-Contradiction') or empty []"],
  "reasoning": "Brief justification."
}
}

Output RAW JSON only.
```

## C.3. Metrics for Predictive Reasoning

For tasks involving natural language outputs, we assess performance using a combination of numerical precision and logical coherence.

**Normalized Mean Absolute Error (NMAE).** To compare regression performance across datasets with vastly different scales (e.g., percentage rates vs. financial volumes), we normalize the error by the full value range of the dataset. For a predicted value $\hat{y}$ and ground truth $y$:

$$\text{NMAE} = \frac{|y - \hat{y}|}{\max(Y) - \min(Y)} \tag{4}$$

```
// Single-Point Prediction (Regression)
{
  "task_type": "regression",
  "prediction_payload": {
    "predicted_value": 10535.0,
    "predicted_interval": [9500.0, 11500.0],
    "proof_quote_value": "Predicted Amount: $10,535",
    "proof_quote_interval": "Most Likely Range: $9,500 - $11,500"
  }
}

// Decision Making (Classification)
{
  "task_type": "classification",
  "scenarios_extraction": {
    "001": {
      "predicted_category": "good",
      "proof_quote": "Apple 1967.0 would be the better choice..."
    },
    "002": {
      "predicted_category": "bad",
      "proof_quote": "Apple 3037.0 is more likely to be the 'bad' apple."
    }
  },
  "final_decision_extraction": {
    "predicted_winner_id": "001",
    "proof_quote": "For your demo, Apple 1967.0 would be the better choice..."
  }
}

// Treatment Effect Analysis (Regression Trend)
{
  "task_type": "regression",
  "scenario_002_extraction": { // The counterfactual scenario
    "predicted_value": 4978.34,
    "predicted_interval": null,
    "proof_quote_value": "Predicted charges: **$4,978.34**"
    "proof_quote_interval": null
  },
  "trend_extraction": {
    "predicted_trend": "higher",
    "proof_quote": "Moving... will significantly INCREASE your medical charges."
  }
}
```

*Figure 8.* **Standardized JSON Schemas for Predictive Reasoning Tasks.** The Judge extracts structured payloads containing point estimates, intervals, and verbatim proof quotes. For B2 and B3, predictions are extracted per scenario to verify comparative reasoning.

where $\max(Y)$ and $\min(Y)$ represent the maximum and minimum observed values of the target column in the historical dataset. If the range is negligible, we default to the absolute magnitude of the ground truth.

**Interval Intersection over Union (IoU) and Width Penalty.** For probabilistic predictions where models output a confidence interval $[\hat{y}_{min}, \hat{y}_{max}]$, we calculate the Intersection over Union (IoU) with a constructed ground truth region. Since historical records typically provide single-point outcomes, we define the ground truth interval as $[y - 0.5\sigma, y + 0.5\sigma]$, where $\sigma$ is the standard deviation of the target variable derived from the dataset statistics. To prevent models from gaming the metric by predicting overly broad ranges (e.g., $[-\infty, \infty]$), we apply an exponential **Width Penalty**:

$$P_{\text{width}} = \exp\left(-\alpha \cdot \max\left(0, \frac{W_{\text{pred}}}{W_{\text{gt}}} - \kappa\right)\right) \tag{5}$$

where $W_{\text{pred}}$ is the predicted width, $W_{\text{gt}} = \sigma$ is the reference width, and $\kappa = 2.0$ is the tolerance threshold. This ensures that precision is not sacrificed for coverage.

**Logic Score.** Beyond accuracy, we evaluate the reasoning process itself. The Judge LLM assigns a scalar score (0-5) based on a rubric that penalizes logical fallacies such as *Self-Contradiction* (text conflicts with prediction), *Circular Reasoning*, and *False Causality* (linking irrelevant IDs to outcomes).

*Table 14.* **Judge Robustness Across Model Families.** We rescore the same outputs with a GPT-family judge and report task-score differences relative to the DeepSeek judge.

| Setting | DS Judge | GPT Judge | Gap |
|---------|----------|-----------|------|
| SP, text | 0.5696 | 0.4934 | -0.0763 |
| SP, tool | 0.5901 | 0.5147 | -0.0754 |
| DM, text | 0.5699 | 0.5780 | +0.0081 |
| DM, tool | 0.5672 | 0.5618 | -0.0054 |
| TE, text | 0.5524 | 0.5905 | +0.0381 |
| TE, tool | 0.6095 | 0.6810 | +0.0714 |
| Overall | 0.5763 | 0.5502 | -0.0262 |

*Table 15.* **Manual Audit of Judge Extraction.** Human annotators checked whether extracted structured values faithfully matched the model output. The only mismatch occurred because the response ambiguously provided multiple options, while the strict judge selected only one classification label. The two judge families also agree on the low-logic flag in 95.1% of cases with logic score below 0.4.

| Task | Audited Cases | Extraction Agreement |
|------|---------------|----------------------|
| Single-Point Prediction | 24 | 100.0% |
| Decision Making | 24 | 95.8% |
| Treatment Effect Analysis | 24 | 100.0% |
| Overall | 72 | 98.6% |

## C.4. Metrics for Structured Ranking

For the Ranking and Filtering task, which outputs structured CSV files, we employ deterministic set-based and rank-aware metrics.

**Set Retrieval Metrics.** We treat the filtering task as a retrieval problem. Let $\mathcal{S}_{gt}$ be the set of ground truth candidates satisfying the implicit conditions, and $\mathcal{S}_{pred}$ be the set of candidates returned by the model. We calculate **Precision**, **Recall**, and **F1 Score** to measure the model's ability to strictly adhere to feature constraints.

**Rank-Aware Metrics.** For queries requiring prioritized lists (e.g., "top 10 highest risk"), we evaluate the ordering quality using **NDCG@k** (Normalized Discounted Cumulative Gain). This metric rewards models for placing high-relevance items at the top of the list, discounting correct items that appear lower down.

**Kendall's Tau ($\tau$).** To assess the model's ability to capture the correct relative order of candidates—even if specific values are imprecise—we compute Kendall's Rank Correlation Coefficient ($\tau$) between the predicted ranking and the ground truth ranking of the matched pairs.

**Batch NMAE.** To verify that the model is performing genuine batch inference rather than simple sorting, we calculate the NMAE averaged exclusively over the correctly retrieved (matched) candidates. This decouples the retrieval performance from the predictive precision.

## D. Extended Experimental Results

This section presents a granular analysis of model performance across three dimensions. First, we examine the quantitative precision and ranking quality in the structured Ranking and Filtering task. We analyze the structural integrity of the generated outputs, highlighting the trade-offs between reasoning depth and format compliance. Finally, we investigate the agentic behaviors of models by profiling their preference for predictive tools across different task types and information settings.

### D.1. Performance on Ranking and Filtering

In this section, we provide a granular analysis of the Ranking and Filtering task. Unlike the natural language reasoning tasks, this scenario requires models to act as data processors, generating structured CSV outputs that satisfy complex filtering conditions and predictive ranking criteria. We evaluate performance across three dimensions: predictive precision, the impact of enhanced schema context, and structural output integrity.

*Table 16.* **Metric Sensitivity on Single-Point Prediction.** We sweep point weight in {0.2,0.3,...,0.8}, width factor in {1.5,2.0,3.0}, and width decay in {0.5,1.0,2.0}.

| Statistic | Text | Tool |
|---|---|---|
| Top-1 unchanged across all profiles | 1.0000 | 1.0000 |
| Mean Spearman agreement vs. default | 0.9960 | 0.9913 |
| Mean pairwise order agreement | 0.9900 | 0.9833 |

### D.1.1. CORE PREDICTIVE PERFORMANCE

We first analyze the performance of General LLMs and Reasoning-Enhanced Models in the standard setting without auxiliary schema descriptions. Table 17 details the metrics for regression ranking and classification filtering.

In the context of regression-based ranking, Gemini 3 Flash and DeepSeek-V3.2-Instruct demonstrate the highest consistency. Gemini 3 Flash achieves the leading NDCG score of 0.5046 and the highest Kendall's $\tau$ of 0.2740, indicating superior capability in ordering high-value candidates correctly relative to the ground truth. DeepSeek-V3.2-Instruct closely follows in ranking quality and distinguishes itself with the highest absolute precision, achieving the lowest NMAE of 0.2639 and a median NMAE of 0.0448. Claude Sonnet 4.5 exhibits a strong retrieval capability with a Recall of 0.5242, comparable to the top performers, but records a higher NMAE (0.3149), suggesting that while it effectively identifies the correct set of candidates, its estimation of their specific attribute values is less precise.

For categorical filtering tasks, Gemini 3 Flash maintains the lead with an F1 score of 0.5751, followed by Claude Sonnet 4.5 at 0.5456. In contrast, the Tabular Specialists, specifically TableLLM-8B and TableLLM-13B, fail to adapt to this complex agentic setting. Their near-zero scores across all metrics highlight the significant gap between domain-specific tuning on short contexts and the long-context reasoning required by TopBench.

*Table 17.* **Core Performance Metrics for Ranking and Filtering.** We report detailed metrics for Regression (Recall, NDCG, NMAE, Kendall's $\tau$) and Classification (F1). *Reg NMAE Med* denotes the median Normalized Mean Absolute Error, providing a robust measure of error unaffected by outliers.

| Model | Regression Metrics | | | | | | Classification Metrics | |
|---|---|---|---|---|---|---|---|---|
| | Recall | Recall (Med) | NDCG | NMAE | NMAE (Med) | Kendall | F1 | Recall |
| **Gemini 3 Flash** | 0.5331 | 0.6000 | **0.5046** | 0.3031 | 0.0489 | **0.2740** | **0.5751** | **0.6787** |
| **DeepSeek-V3.2-Instruct** | **0.5393** | 0.6000 | 0.5045 | **0.2639** | **0.0448** | 0.2148 | 0.4838 | 0.5671 |
| **Claude Sonnet 4.5** | 0.5242 | **0.6125** | 0.4964 | 0.3149 | 0.0517 | 0.2351 | 0.5456 | 0.6694 |
| **GPT-5.2** | 0.4561 | 0.4000 | 0.4149 | 0.4058 | 0.1028 | 0.2031 | 0.3825 | 0.4372 |
| **DeepSeek-V3.2-Thinking** | 0.4977 | 0.6000 | 0.4624 | 0.3527 | 0.0697 | 0.1802 | 0.3842 | 0.4584 |
| **Qwen3-Thinking** | 0.4289 | 0.4000 | 0.4096 | 0.4550 | 0.1465 | 0.1838 | 0.4191 | 0.5186 |
| **Qwen3-Instruct** | 0.3069 | 0.0000 | 0.2791 | 0.6057 | 1.0000 | 0.0557 | 0.2782 | 0.4402 |
| TableLLM-8B | 0.0028 | 0.0000 | 0.0040 | 0.9951 | 1.0000 | 0.0000 | 0.0001 | 0.0171 |
| TableLLM-13B | 0.0000 | 0.0000 | 0.0000 | 1.0000 | 1.0000 | 0.0000 | 0.0000 | 0.0000 |

### D.1.2. IMPACT OF ENHANCED SCHEMA CONTEXT

We investigate whether providing rich schema metadata (column definitions and value distributions) improves performance. Table 18 contrasts the Standard models with their With Info Mode variants.

The provision of metadata yields significant gains for models with weaker initial performance. Qwen3-Instruct observes a substantial boost, with Regression Recall increasing from 0.3069 to 0.3693 and Median NMAE improving markedly from 1.0 to 0.8309. Similarly, GPT-5.2 sees consistent improvements, particularly in classification tasks where its F1 score rises from 0.3825 to 0.4760. Conversely, top-tier frontier models such as Gemini 3 Flash and DeepSeek-V3.2-Instruct show minimal performance variance or diminishing returns. This stability suggests that these advanced models possess robust internal capabilities to infer schema semantics directly from raw tabular data, rendering explicit metadata less critical for their reasoning processes compared to smaller models.

*Table 18.* **Impact of Contextual Schema Information.** Comparison of key metrics between standard models and models provided with auxiliary schema info (`w/ Info`). Metrics shown are Regression Recall, Regression NMAE, and Classification F1.

| Base Model | Standard Mode | | | With Info Mode | | |
|---|---|---|---|---|---|---|
| | Reg Recall | Reg NMAE | Cls F1 | Reg Recall | Reg NMAE | Cls F1 |
| **DeepSeek-V3.2-Instruct** | 0.5393 | 0.2639 | 0.4838 | **0.5490** | **0.2398** | **0.5462** |
| **Gemini 3 Flash** | **0.5331** | **0.3031** | **0.5751** | 0.5291 | 0.3072 | 0.5659 |
| **GPT-5.2** | 0.4561 | 0.4058 | 0.3825 | **0.4851** | **0.3542** | **0.4760** |
| **Qwen3-Instruct** | 0.3069 | 0.6057 | 0.2782 | **0.3693** | **0.5466** | **0.3339** |

### D.1.3. STRUCTURAL INTEGRITY AND COMPLIANCE ANALYSIS

Given that Ranking tasks necessitates the generation of valid CSV files, we evaluate the structural integrity of the output in Table 19.

A notable trade-off emerges between reasoning depth and format compliance. DeepSeek-V3.2-Thinking, while competitive in predictive metrics, exhibits a high CSV Parse Error Rate of 16.82% and a corresponding Missing Columns Rate. This suggests that models optimized for extensive Chain-of-Thought reasoning may struggle to separate internal monologue from the strict formatting requirements of structured file generation. In contrast, Gemini 3 Flash achieves flawless structural integrity with 0.00% error rates across all categories. Regarding logical adherence, the Filter Compliance Rate remains consistently high ($> 98\%$) across all generalist models, confirming that the primary challenge in TOPBENCH is the implicit prediction of target variables rather than the execution of explicit filtering instructions.

*Table 19.* **Structural Integrity and Error Analysis.** We report the rate of CSV parsing failures, schema hallucinations (Extra/Missing Columns), and the logical Filter Compliance Rate. Lower error rates and higher compliance indicate better instruction following.

| Model | CSV Parse Error | Missing Cols | Extra Cols | Empty Result | Filter Compliance |
|---|---|---|---|---|---|
| **Gemini 3 Flash** | 0.00% | 0.00% | 0.00% | 0.00% | **100.00%** |
| **DeepSeek-V3.2-Thinking** | 16.82% | 16.82% | 0.47% | 0.00% | **100.00%** |
| **GPT-5.2** | 0.00% | 0.00% | 0.47% | 0.00% | 99.53% |
| **Qwen3-Thinking** | 0.00% | 0.93% | 0.93% | 0.00% | 99.53% |
| **Claude Sonnet 4.5** | 1.40% | 1.40% | 0.47% | 0.47% | 99.07% |
| **Qwen3-Instruct** | 1.40% | 5.14% | 0.93% | 0.47% | 99.07% |
| **DeepSeek-V3.2-Instruct** | 0.47% | 0.47% | 0.93% | 0.00% | 98.60% |
| *TableLLM-8B* | 45.33% | 47.66% | 6.54% | 0.00% | 98.13% |

### D.2. Analysis of Predictive Tool Usage Behaviors

We provide a granular analysis of how different Large Language Models (LLMs) select and utilize computational tools across the four predictive tasks. By comparing the tool invocation distributions in the standard setting against the semantic-enhanced setting, we isolate the impact of intent recognition on modeling strategy.

**Baseline Tool Usage Patterns.** In the standard setting, distinct behavioral profiles emerge among the evaluated models. As illustrated in Figures 9 through 15, DeepSeek-V3.2-Instruct consistently exhibits a strong preference for rigorous predictive modeling, employing algorithms such as Random Forest in over 50% of cases across Single Point Prediction and Treatment Effect Analysis tasks. This suggests an intrinsic alignment with the predictive nature of the queries. In contrast, Qwen3-Instruct demonstrates a marked tendency towards heuristic solutions, predominantly relying on `pandas` for data filtering or performing direct arithmetic calculations. This behavior indicates that without explicit guidance, the model often misinterprets the implicit predictive intent as a retrieval or simple aggregation task. Notably, in the Ranking and Filtering task (Figure 15), we observe a universal increase in the complexity of chosen algorithms, with models frequently deploying gradient boosting frameworks to handle the batch scoring requirements.

**Impact of Semantic Disambiguation.** The introduction of explicit semantic information—specifying target columns and task types—triggers a significant shift in agentic behavior. Comparing the baseline distributions with their enhanced

counterparts (Figures 10 to 16) reveals a convergence towards formal statistical modeling. The most dramatic transformation is observed in Qwen3-Instruct. In the Single Point Prediction task, its utilization of machine learning libraries surges, effectively bridging the strategic gap with DeepSeek and GPT-5.2. This behavioral correction confirms that the model's prior reliance on simple heuristics stemmed primarily from intent ambiguity rather than a lack of coding capability. Furthermore, for the Decision Making task, while the shift in tool usage is evident, the accompanying performance gains are less pronounced, highlighting that correct tool invocation is a necessary but not sufficient condition for resolving fine-grained trade-offs.

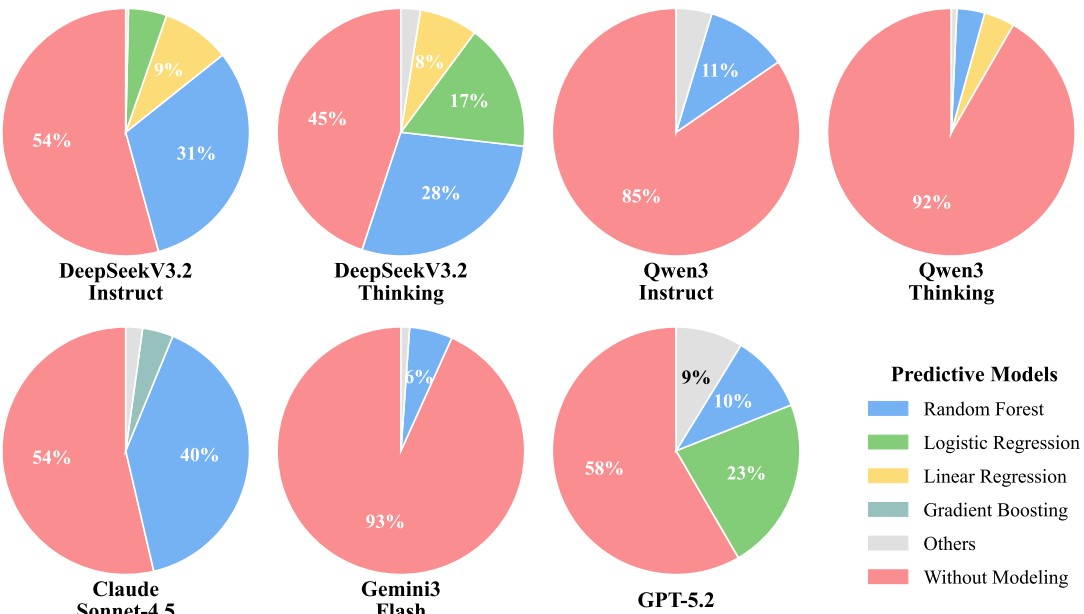

*Figure 9.* **Tool Usage Distribution for Single Point Prediction (Standard).** DeepSeek actively employs predictive models, while Qwen3 relies heavily on non-modeling approaches.

**Integrity of the Modeling Pipeline.** Merely invoking a machine learning library does not guarantee a valid prediction; the raw tabular data must be rigorously prepared. We further investigate the completeness of the generated code by measuring the rate of data preprocessing—specifically, the implementation of categorical encoding or missing value imputation—given that a predictive model was instantiated. As shown in Figure 17, there is a strong correlation between model capability and pipeline integrity. Advanced agents not only select the correct algorithms but also autonomously recognize the need to transform raw string-based features into numerical formats compatible with standard libraries. In contrast, despite correctly identifying the need for prediction, weaker models frequently attempt to feed raw data directly into regressors. This observation highlights that true agentic intelligence extends beyond simple intent recognition to the comprehensive emulation of a data scientist's workflow, encompassing both algorithm selection and essential data hygiene.

# E. Qualitative Case Studies and Error Analysis

In this section, we provide a qualitative examination of model behaviors, contrasting successful execution patterns in the agentic framework with typical failure modes observed in the text-based setting. We begin by analyzing the correlation between response verbosity and reasoning effectiveness across different model families.

### E.1. Analysis of Response Length and Verbosity

We investigate the distribution of generated token counts across different inference modes to understand how tool availability influences model expressiveness. Figure 18 illustrates the average response length for each model, comparing the Text-Based Reasoning mode (No Tool) against the Agentic Workflow (With Tool).

The data reveals a counter-intuitive phenomenon where "verbosity" often correlates inversely with "reasoning quality," a trend that varies significantly across model types. In the Text-Based setting, open-weights models such as Qwen3-Instruct and

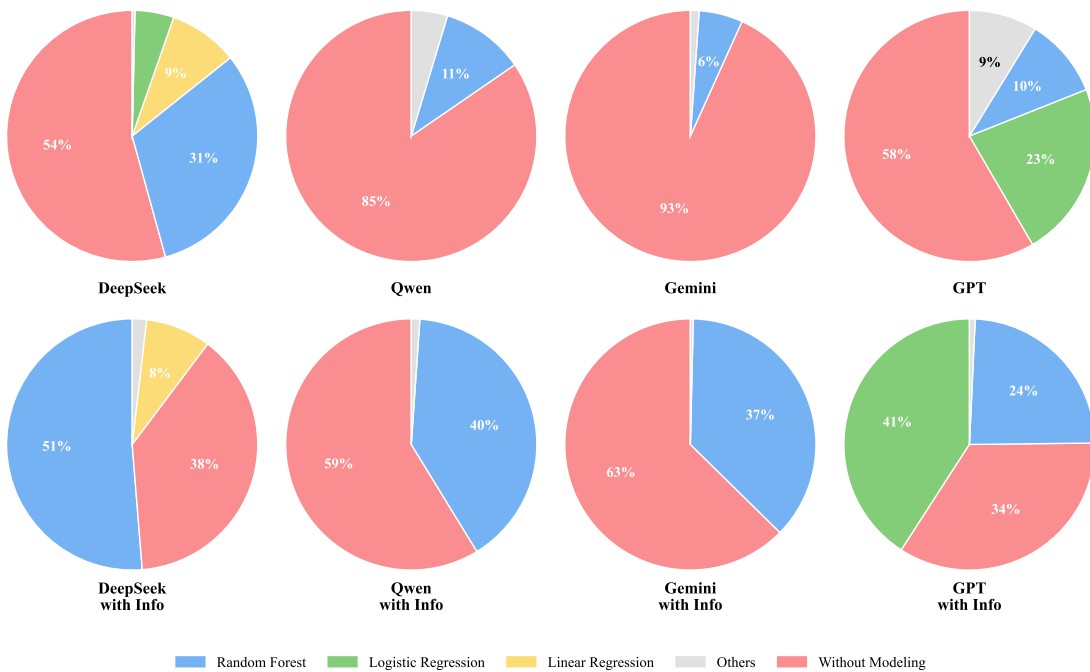

*Figure 10.* **Tool Usage Distribution for Single Point Prediction (With Semantic Info).** The addition of semantic metadata prompts a significant increase in modeling frequency for Qwen3.

DeepSeek-V3.2 exhibits the most inflated response lengths. Lacking the internal capacity for precise arithmetic, these models frequently resort to generating extensive, convoluted reasoning chains to justify heuristic estimates. This "hallucination by verbosity"suggests an attempt to simulate logical depth through length. However, upon enabling the Agentic Workflow, these specific models show the most dramatic reduction in token usage. By offloading complex calculations to the interpreter, they transition from verbose rhetoricians to efficient dispatchers, converging towards the naturally more concise output patterns observed in proprietary models like GPT-5.2. This indicates that for open-weights models, tool access serves as a critical regularizer, effectively pruning unnecessary textual generation in favor of deterministic executable logic.

### E.2. The Retrieval Trap in Text-Based Reasoning

A pervasive limitation observed across all evaluated models in the text-based setting is the tendency to reduce predictive problems to heuristic retrieval tasks. Lacking the computational tools to fit a statistical function $f(x) \rightarrow y$, models rely on identifying "nearest neighbors"within the truncated context window to infer the target value.

**Heuristic Range Estimation vs. Model Bias.** As illustrated in Figure 19, when presented with a query profile in the text-only mode, the model locates historical rows with similar feature values (e.g., matching region or age). Instead of producing a precise point estimate, it aggregates these retrieved values to construct a fuzzy confidence interval (e.g., "between \$9,630 and \$10,000"). While this approach provides a conservative estimate, it lacks the mathematical precision to capture complex non-linear dependencies.

In contrast, Figure 20 demonstrates how the Agentic Workflow enables the model to explicitly train a regressor. However, **methodological superiority does not guarantee numerical precision.** While the agent successfully transitions from retrieval to modeling, the predicted range (\$13,500–\$14,000) still deviates significantly from the ground truth (\$9,391). This error stems from the model's tendency to default to simple algorithms (e.g., unoptimized Linear Regression) within the zero-shot sandbox, which often underfit complex real-world distributions characterized by non-linear interaction terms (e.g., the interplay between BMI and age). Thus, while the Agentic framework corrects the *process*, the *accuracy* remains bound by the sophistication of the generated code.

**Superficial Feature Matching in Decision Making.** This retrieval-based bias is equally detrimental in decision-making tasks. Figure 21 shows that without tools, models attempt to predict the better candidate by searching for historically similar

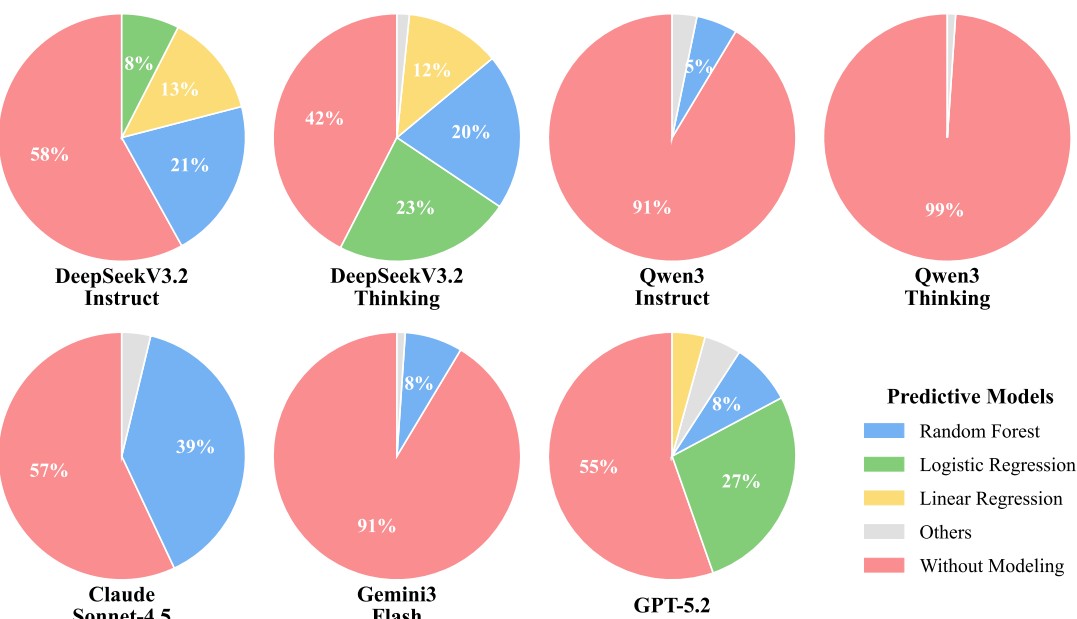

*Figure 11.* **Tool Usage Distribution for Decision Making (Standard).** Models show a mixed strategy, balancing between comparison logic and predictive modeling.

instances. However, this often leads to "superficial feature matching,"where the model latches onto salient but non-causal features to justify its choice. Conversely, the Agentic approach shown in Figure 22 allows the model to quantify the expected outcome for both candidates independently using a trained predictor, transforming the decision from a qualitative guess into a quantitative comparison $\text{argmax}(y_{pred})$, though subject to the same regression precision limitations noted above.

### E.3. Failure Mode: The Exhaustive Retrieval Loop

Beyond the general tendency for retrieval, we observe a specific, catastrophic failure mode in reasoning-enhanced models, most notably Qwen3-Thinking. We term this the "Exhaustive Retrieval Loop."

As depicted in Figure 23, this error occurs when the model fundamentally misinterprets the implicit prediction task as a strict database lookup. Faced with a query describing an unobserved profile (e.g., a hypothetical fruit or patient), the model operates under the false assumption that an exact match exists within the historical log. Consequently, it initiates a brute-force linear search, mechanically iterating through the serialized table string row by row. The reasoning trace degenerates into a repetitive cycle of element-wise verification (e.g., "Checking entry 1... mismatch. Checking entry 2... mismatch."), consuming the entire context window. Since the query target is a future state rather than a historical record, this search is inherently futile. The model eventually exhausts its maximum token limit or hallucinates a match to break the loop. This behavior highlights a critical cognitive gap: despite strong logical capabilities, the model lacks the tabular awareness to distinguish between querying the past and modeling the future.

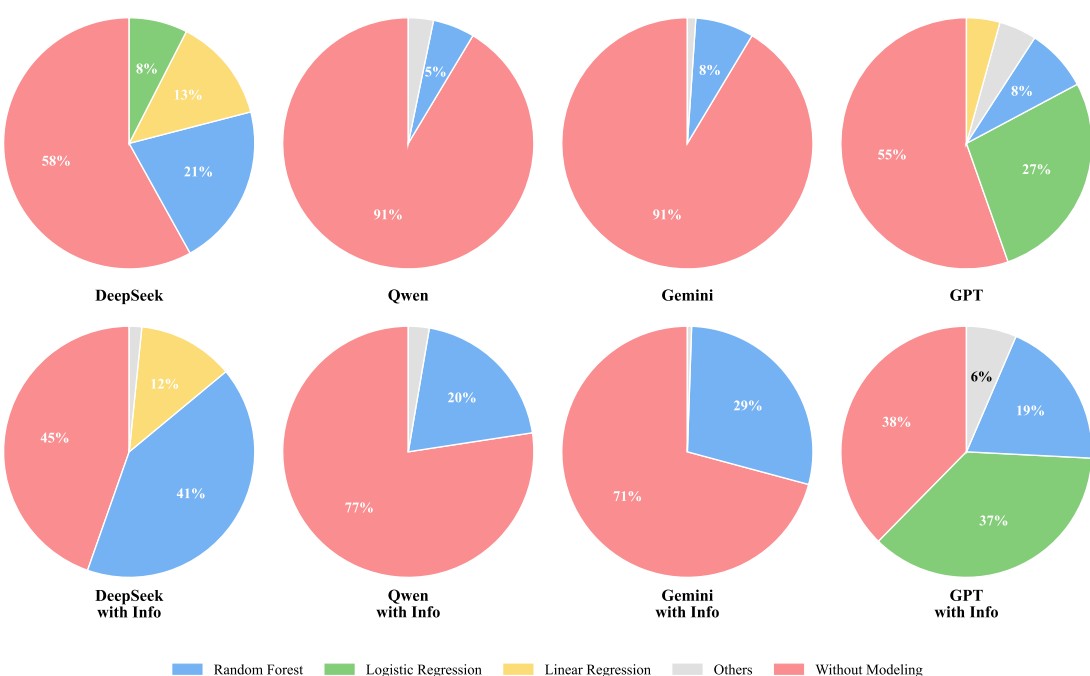

*Figure 12.* **Tool Usage Distribution for Decision Making (With Semantic Info).** Explicit task definition encourages models to adopt more formal comparative analysis techniques.

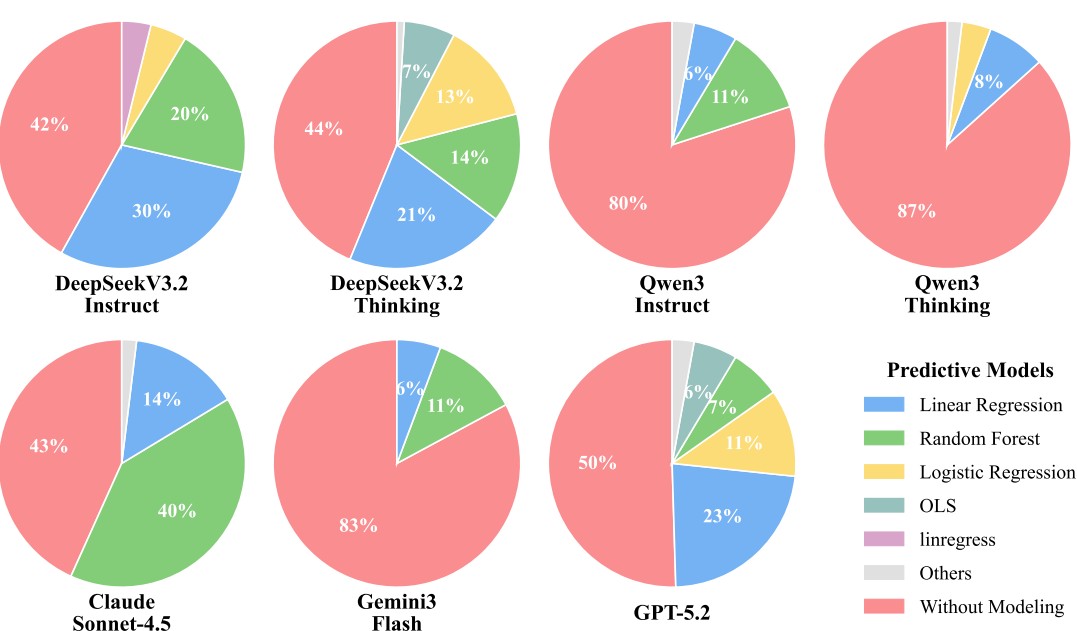

*Figure 13.* **Tool Usage Distribution for Treatment Effect Analysis (Standard).** Causal reasoning scenarios drive a higher baseline usage of regression models across all agents.

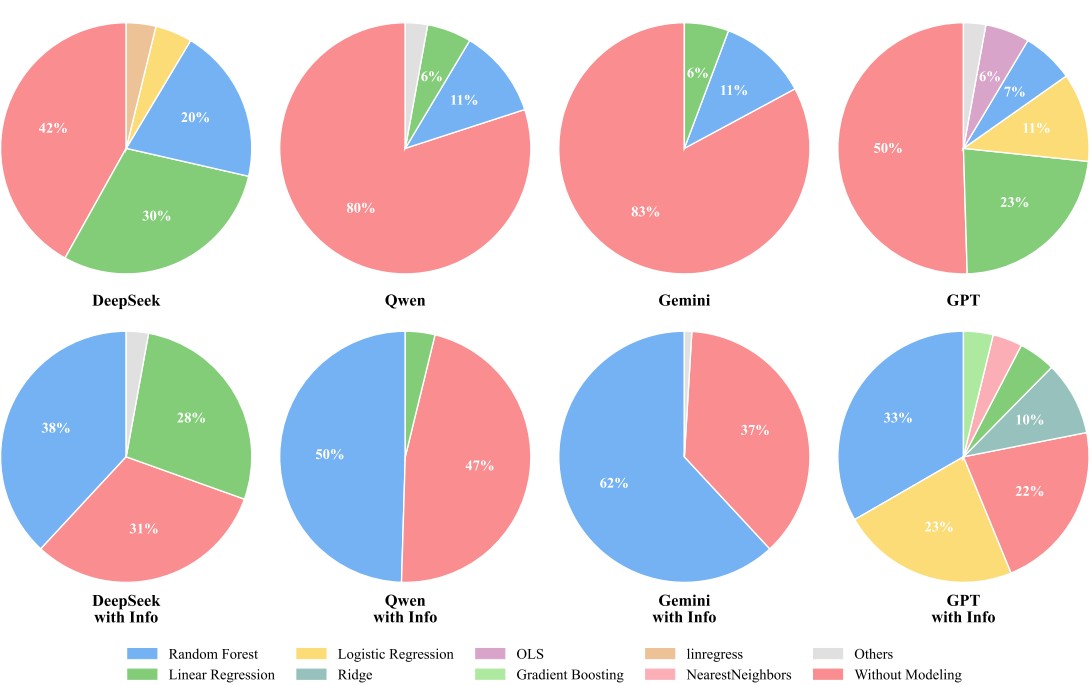

*Figure 14.* **Tool Usage Distribution for Treatment Effect Analysis (With Semantic Info).** Enhanced context further solidifies the preference for causal inference methods over heuristic estimation.

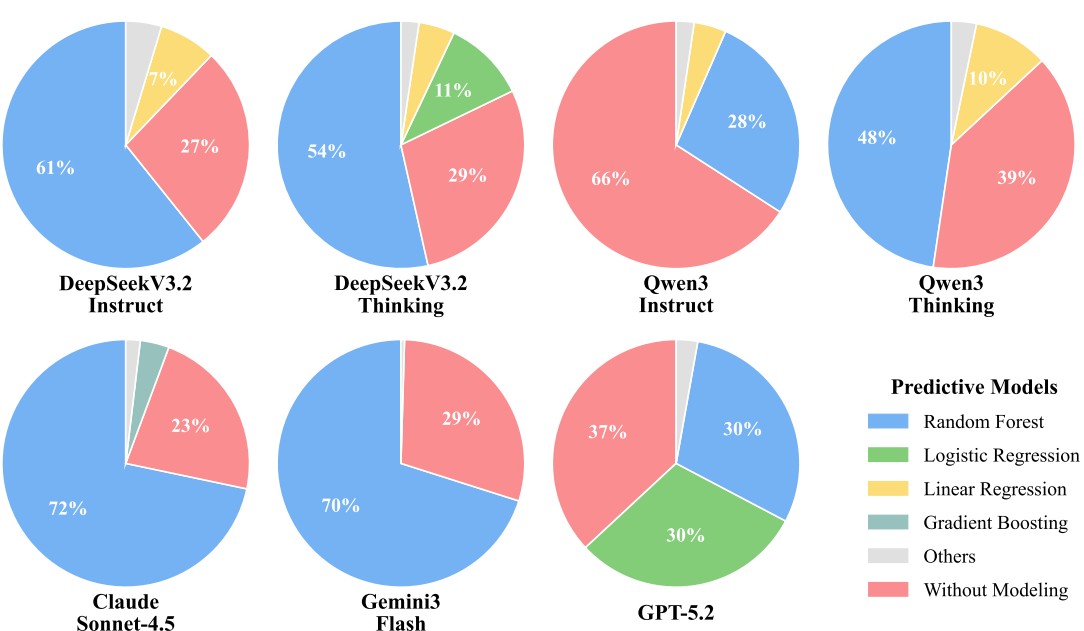

*Figure 15.* **Tool Usage Distribution for Ranking and Filtering (Standard).** The complexity of batch processing naturally leads to a higher adoption of robust algorithms like Random Forest.

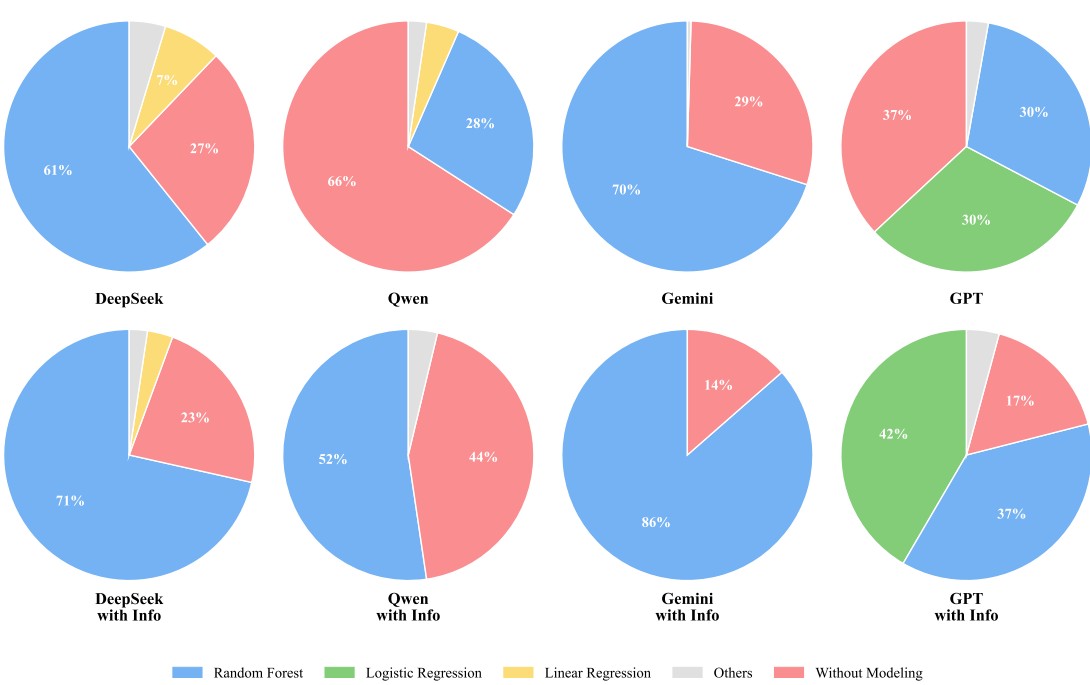

*Figure 16.* **Tool Usage Distribution for Ranking and Filtering (With Semantic Info).** Semantic clarity assists models in selecting more appropriate feature sets for batch ranking algorithms.

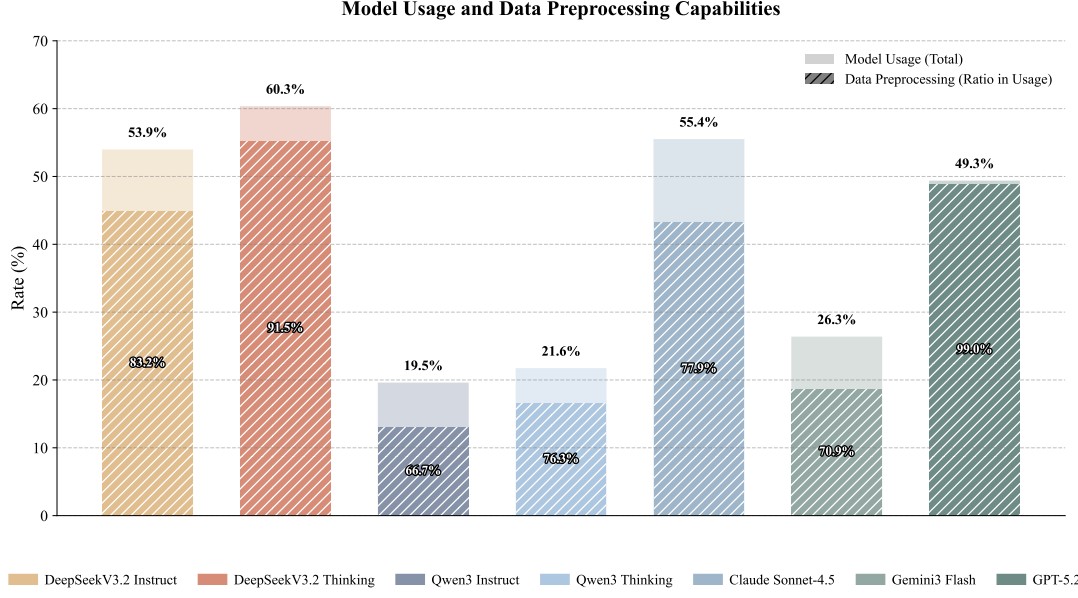

*Figure 17.* **Ratio of Data Preprocessing in Predictive Workflows.** The chart quantifies the conditional probability that a model implements necessary feature engineering steps (e.g., encoding, imputation) when employing machine learning algorithms. Higher ratios indicate more robust and executable code generation.

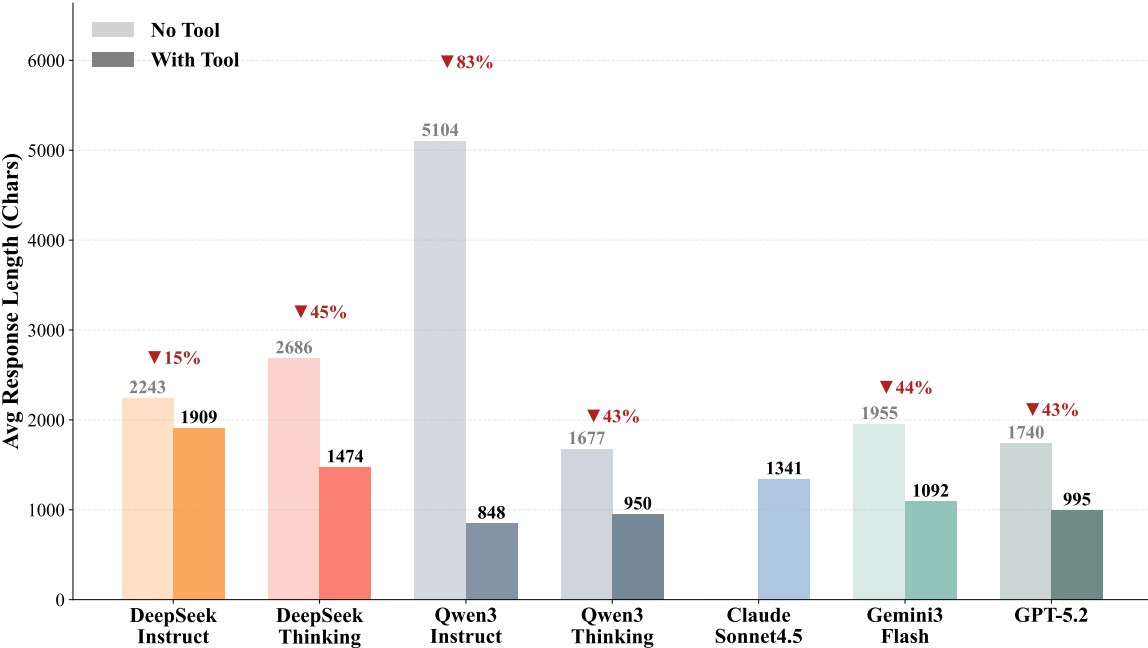

*Figure 18.* **Comparison of Average Response Lengths.** We contrast the token usage between Text-Based Reasoning (No Tool), where models rely solely on internal parameters, and the Agentic Workflow (With Tool), where models utilize Python execution. The lengths reported include the generated reasoning text and code blocks.

## Single-Point Prediction
## Text-Based

**Context:**
age,sex,bmi,children,smoker,region,charges
23,male,31.73,3,yes,northeast,36189.1017
19,male,28.7,0,no,southwest,1253.936
19,male,20.9,1,no,southwest,1832.094
19,male,33.1,0,no,southwest,23082.95533
18,male,31.68,2,yes,southeast,34303.1672
48,female,28.88,1,no,northwest,9249.4952
57,male,28.975,0,yes,northeast,27218.43725
25,male,24.985,2,no,northeast,23241.47453
31,female,38.095,1,yes,northeast,58571.07448

**Ground Truth:** "charges": 9391.346

**Query:** Just had my annual checkup, and the doctor reviewed everything with me. So, being a 51-year-old man living in the southwest, he noted my BMI is 39.7. I've got one child on my insurance plan, and while I'm not a smoker, he did express some concern about my weight. With all that in mind, what kind of medical insurance bill should I be preparing for this year?

**Response From DeepseekV3.2-Instruct:** Based on the historical insurance data you provided, for a 51-year-old male in the southwest with a BMI of 39.7, one child, and who is a non-smoker, the estimated annual medical insurance charges would be approximately **$9,630 to $10,000**.

Here's a breakdown of the key factors influencing this estimate:

*   **Age (51):** This is a significant factor, as premiums generally increase with age.
*   **BMI (39.7):** This falls into the obese category (BMI ≥ 30), which typically leads to higher premiums due to associated health risks.
*   **Region (Southwest):** Costs can vary by region, and the southwest shows premiums in this range for similar profiles.
*   **Smoker Status (Non-Smoker):** This keeps your costs lower than if you were a smoker.
*   **Children (1):** Adding one dependent child increases the premium.

**Important Disclaimer:** This is an estimate based on historical data patterns. Your actual premium will be determined by the specific insurance plan you choose (e.g., deductible, co-pays, network), your exact location within the southwest, the insurance provider, and your complete medical history.

**Recommended Next Steps:**
1.  **Get Official Quotes:** Contact insurance providers directly or use the Health Insurance Marketplace for personalized quotes.
2.  **Consider Plan Details:** Look beyond the premium at deductibles, out-of-pocket maximums, and covered services.
3.  **Discuss with Your Doctor:** His concern about weight is a health prompt. Addressing that could improve your long-term health and potentially lower future insurance costs.

*Figure 19.* **Single-Point Prediction (Text-Based).** The model resorts to retrieving similar historical rows and estimating a vague range based on neighbor values. While semantically plausible, this approach fails to capture the precise, non-linear mapping required for the ground truth target.

## Single-Point Prediction
## Agentic-Workflow

Ground Truth: "charges": 9391.346

**Query:** Just had my annual checkup, and the doctor reviewed everything with me. So, being a 51-year-old man living in the southwest, he noted my BMI is 39.7. I've got one child on my insurance plan, and while I'm not a smoker, he did express some concern about my weight. With all that in mind, what kind of medical insurance bill should I be preparing for this year?

**Context:**
age,sex,bmi,children,smoker,region,charges
23,male,31.73,3,yes,northeast,36189.1017
19,male,28.7,0,no,southwest,1253.936
19,male,20.9,1,no,southwest,1832.094
19,male,33.1,0,no,southwest,23082.95533
18,male,31.68,2,yes,southeast,34303.1672
48,female,28.88,1,no,northwest,9249.4952
57,male,28.975,0,yes,northeast,27218.43725
25,male,24.985,2,no,northeast,23241.47453
31,female,38.095,1,yes,northeast,58571.07448

**Response From DeepseekV3.2-Instruct:**
[Code] from sklearn.linear_model import LinearRegression
from sklearn.preprocessing import LabelEncoder
Based on my analysis of the insurance data, here's what you should prepare for in terms of medical insurance costs:
## **Predicted Annual Insurance Bill: $13,500 - $14,000**

### **Key Factors Driving Your Costs:**

1. **Age (51 years)**: Adds about $255 per year of age
2. **High BMI (39.7)**: Your BMI is in the 92nd percentile, adding approximately $330 per BMI unit
3. **Non-smoker status**: This is actually saving you about **$23,400 per year** compared to smokers!
4. **One dependent child**: Adds about $519 to your premium
5. **Southwest region**: Slightly lower costs than some other regions

### **Comparison Data:**

- **Most similar profiles** (men 45-57, southwest, non-smokers, 1 child, high BMI): $8,068 - $11,576
- **Your predicted range**: $13,500 - $14,000 (slightly higher due to your specific age and exact BMI)
- **All non-smokers average**: $8,452
- **All smokers average**: $31,876
......

*Figure 20.* **Single-Point Prediction (Agentic Workflow).** Leveraging the sandbox, the model trains a Linear Regression model to generate a numerical estimate. Note that while the *approach* is correct (modeling vs. retrieval), the *result* ($13.5k) still deviates from the ground truth ($9.3k), highlighting the limitation of simple default algorithms in zero-shot code generation.

**Decision Making
Text-Based**

**Ground Truth:**
"charges": "1725.5523"
"charges": "1720.3537"
"charges": "1727.54"

**Context:**
age,sex,bmi,children,smoker,region,charges
23,male,31.73,3,yes,northeast,36189.1017
19,male,28.7,0,no,southwest,1253.936
19,male,20.9,1,no,southwest,1832.094
19,male,33.1,0,no,southwest,23082.95533
18,male,31.68,2,yes,southeast,34303.1672
48,female,28.88,1,no,northwest,9249.4952
57,male,28.975,0,yes,northeast,27218.43725
25,male,24.985,2,no,northeast,23241.47453
31,female,38.095,1,yes,northeast,58571.07448

**Query:** I'm trying to help my younger cousin get a sense of his potential health insurance costs as he's looking at plans... The first person is an 18-year-old man from the Southeast. He's not a smoker, has a Body Mass Index of 33.77, and has one child covered on his plan. The second record is also... Finally, the third option... With all these similar but slightly different profiles, I'm a bit stuck on figuring out which one of these three would likely have the lowest medical costs billed by insurance. Can you help me sort that out?

**Response From DeepseekV3.2-Instruct:** Let's break this down. You have three 18-year-old male non-smokers from the Southeast, each with one child, and the only difference is their BMI:

1. **Person A**: BMI = 33.77
2. **Person B**: BMI = 30.03
3. **Person C**: BMI = 35.2

From the dataset, I can find records that closely match these profiles to estimate their insurance charges.
### **Matching records from the dataset**
......

### **General trend from data**
Looking at 18-year-old males in Southeast, non-smoker, 0 children (since 1-child records are rare for this age in dataset):
......
### **Estimated order of cost (lowest to highest)**
1. **Person B (BMI 30.03)** — lowest BMI → lowest base charge.
2. **Person A (BMI 33.77)** — middle BMI → middle charge.
3. **Person C (BMI 35.2)** — highest BMI → highest charge.

**Answer**:
**Person B** (BMI 30.03) would likely have the lowest medical costs, followed by Person A, then Person C

*Figure 21.* **Decision Making (Text-Based).** The model relies on qualitative comparisons with retrieved historical samples. The reasoning is fragile, often basing the decision on superficial feature similarities (e.g., region matches) rather than calculated risk factors.

## Decision Making Agentic-Workflow

**Ground Truth:**
"charges": "1725.5523"
"charges": "1720.3537"
"charges": "1727.54"

**Context:**
age,sex,bmi,children,smoker,region,charges
23,male,31.73,3,yes,northeast,36189.1017
19,male,28.7,0,no,southwest,1253.936
19,male,20.9,1,no,southwest,1832.094
19,male,33.1,0,no,southwest,23082.95533
18,male,31.68,2,yes,southeast,34303.1672
48,female,28.88,1,no,northwest,9249.4952
57,male,28.975,0,yes,northeast,27218.43725
25,male,24.985,2,no,northeast,23241.47453
31,female,38.095,1,yes,northeast,58571.07448

**Query:** I'm trying to help my younger cousin get a sense of his potential health insurance costs as he's looking at plans... The first person is an 18-year-old man from the Southeast. He's not a smoker, has a Body Mass Index of 33.77, and has one child covered on his plan. The second record is also... Finally, the third option... With all these similar but slightly different profiles, I'm a bit stuck on figuring out which one of these three would likely have the lowest medical costs billed by insurance. Can you help me sort that out?

**Response From DeepseekV3.2-Instruct:**
[Code] from sklearn.linear_model import LinearRegression
Based on my analysis of the health insurance data, I can provide you with a clear answer:

## **Summary of Findings**

After analyzing the insurance data and building predictive models, here's what I found for your three scenarios:

### **Predicted Insurance Charges:**
1. **Second person (BMI 30.03)**: $2,211.46
2. **First person (BMI 33.77)**: $2,286.33
3. **Third person (BMI 35.20)**: $2,314.95

......
### **Final Recommendation:**
Your cousin should expect that **lower BMI generally correlates with lower insurance costs**. Among the three scenarios presented, the profile with **BMI 30.03** would likely have the most affordable insurance premiums.

The good news is that all three scenarios are relatively close in predicted costs (within about $100 of each other annually), and all are significantly below the average for non-smokers ($8,451), primarily due to the young age and non-smoking status.

*Figure 22.* **Decision Making (Agentic Workflow).** The model quantifies the decision by predicting exact scores for both candidates. This quantitative comparison enables a more rigorous trade-off analysis compared to the fuzzy logic of the text-based baseline.

## Single-Point Prediction
## Text-Based
## Qwen Error

Ground Truth: "charges": 9391.346

**Query:** Just had my annual checkup, and the doctor reviewed everything with me. So, being a 51-year-old man living in the southwest, he noted my BMI is 39.7. I've got one child on my insurance plan, and while I'm not a smoker, he did express some concern about my weight. With all that in mind, what kind of medical insurance bill should I be preparing for this year?

**Context:**
age,sex,bmi,children,smoker,region,charges
23,male,31.73,3,yes,northeast,36189.1017
19,male,28.7,0,no,southwest,1253.936
19,male,20.9,1,no,southwest,1832.094
19,male,33.1,0,no,southwest,23082.95533
18,male,31.68,2,yes,southeast,34303.1672
48,female,28.88,1,no,northwest,9249.4952
57,male,28.975,0,yes,northeast,27218.43725
25,male,24.985,2,no,northeast,23241.47453
31,female,38.095,1,yes,northeast,58571.07448

**Response From Qwen3-Thinking: No Result**
[Thinking Process / Chain of Thought]:
Let me check if there are any entries with BMI around 39.7, non-smoker, in the southwest region:
60,male,39.9,0,yes,southwest,48173.361 (smoker)
55,male,38.28,0,no,southeast,10226.2842 (different region)
50,male,39.6,0,no,southwest,8457.818 (0 children)

Actually, I found this entry:
51,male,39.7,0,no,southwest,10713.644

This is very close to what we're looking for, except it has 0 children instead of 1.

Let me check if there are any entries with 1 child in the southwest region:
I don't see any exact matches, but let me check more carefully.

Actually, I found this entry:
48,male,37.05,1,yes,southwest,24603.04837
This has 1 child and is in the southwest region, but the person is a smoker (yes).

Let me check if there are any entries with 1 child, non-smoker, in the southwest region:
I don't see any exact matches...

*Figure 23.* **The Exhaustive Retrieval Loop (Qwen3-Thinking).** Misinterpreting the prediction task as a database lookup, the model enters an infinite loop of row-by-row verification. It attempts to find an exact match for a hypothetical profile, eventually exhausting the context window without producing a valid answer.

