# OpenReview forum: "TopBench: A Benchmark for Implicit Predictive Reasoning in Tabular Question Answering"
_ICML.cc/2026/Conference — ICML 2026 regular_

### Official Review · Reviewer_s3Jk · 2026-02-28

**Soundness:** 2
**Presentation:** 3
**Significance:** 1
**Originality:** 1
**Overall Recommendation:** 4
**Confidence:** 4

**Summary:**

The paper introduces a novel benchmark TOPBENCH designed to evaluate Large Language Models (LLMs) on Implicit Predictive Table Question Answering (TQA), a task that goes beyond simple data retrieval to require the inference of unobserved values from historical patterns. The authors present a dataset of 779 samples across four sub-tasks, including single-point prediction, decision making, treatment effect analysis, and complex filtering, which demand that models recognize latent predictive intent and perform reasoning over massive tables. Experimental results indicate that current LLMs often struggle with these tasks, frequently defaulting to basic lookups because they fail to identify the predictive nature of queries. Ultimately, the study highlights that improving performance in this area requires better intent disambiguation and the integration of more sophisticated modeling or reasoning capabilities.

**Compliance With Llm Reviewing Policy:**

Affirmed.

**Final Justification:**

My concern throughout the review process has consistently been that the paper fails to adequately acknowledge and position itself against prior work. This lack of sufficient citation places the paper in an inappropriate context and may consequently mislead readers into overestimating the extent of the authors’ contribution. Nevertheless, in light of the authors’ rebuttal, in which they commit to expanding the related-work section, incorporating strong traditional tabular baselines, and clarifying the precise scope of their contribution, I am willing to increase my score to 4.

**Key Questions For Authors:**

1. It is suggested that the authors provide additional statistics for the benchmark, specifically the distribution of feature counts (i.e., the number of columns) per table.

**Limitations:**

yes

**Strengths And Weaknesses:**

**Strengths**

* The task of tabular prediction is highly significant with a wide range of real-world applications.
* The manuscript is well-structured and easy to understand.
* The authors have designed customized evaluation metrics tailored to different answer types.

**Weaknesses**

* **Literature Review and Contextualization:** The primary concern is that tabular prediction is a well-established research problem [1-2] with numerous existing datasets [3-6]. However, the authors fail to mention these or differentiate their work from prior studies.
* **Baseline Comparisons:** The experiments focus exclusively on Large Language Models (LLMs). The study overlooks traditional algorithms and specialized tabular prediction models, such as XGBoost [7], CatBoost [8], and TabPFNv2 [9].

[1] Deep Neural Networks and Tabular Data: A Survey. IEEE transactions on neural networks and learning systems, 2022.

[2] A Survey on Deep Tabular Learning. ArXiv:2410.

[3] OpenML Benchmarking Suites. ArXiv:1708.

[4] OpenML-CTR23–a curated tabular regression benchmarking suite." AutoML Conference 2023 (Workshop).

[5] When do neural nets outperform boosted trees on tabular data?." Advances in Neural Information Processing Systems 36 (2023).

[6] TabArena: A Living Benchmark for Machine Learning on Tabular Data. The Thirty-ninth Annual Conference on Neural Information Processing Systems Datasets and Benchmarks Track. 2025.

[7] XGBoost: A scalable tree boosting system. Proceedings of the 22nd ACM SIGKDD International Conference on Knowledge Discovery and Data Mining (KDD’16).

[8] Catboost: Unbiased boosting with categorical features. Proceedings of the 31st International Conference on

Advances in Neural Information Processing Systems (NeurIPS’18).

[9] Accurate predictions on small data with a tabular foundation model. Nature, 2025.

---

> ### Author Rebuttal · Authors · 2026-03-30
>
> Thank you for your review and helpful suggestions. We provide our responses below.
>
> >W1: The paper is not well connected to prior tabular prediction work.
>
> We appreciate this point and we will expand the related-work section to cover major tabular-learning surveys, OpenML-style benchmark suites, OpenML-CTR23, TabArena, XGBoost, CatBoost, and TabPFN.
>
> **More importantly**, we will make the task boundary much more explicit. Traditional tabular prediction benchmarks usually assume that the predictive task is already fully specified: the target column is known, the train/test split is given, and the model only needs to map structured features to labels. TOPBENCH studies a **different stage** of the pipeline. In TOPBENCH, the system starts from a natural-language query over a historical table, and must first infer what prediction problem is asking, which target or decision criterion is relevant, and which hypothetical or current instance should be evaluated. Only then can prediction happen.
>
> So our goal is not to replace classical tabular prediction benchmarks. Rather, TOPBENCH is meant to be **complementary** to them: it evaluates the **language-to-prediction** interface that standard tabular benchmarks largely assume away. We will make this distinction explicit much earlier in the revised paper.
>
> ---
>
> >W2: Traditional tabular baselines are missing.
>
> We agree this is an important point, and we have now added stronger non-LLM baselines precisely to separate two sources of difficulty:
>
> 1. the difficulty of tabular prediction itself,
> 2. the difficulty of mapping a natural-language request into the correct prediction task.
>
> We compare four settings below.
>
> | Method | What it isolates | Single-Point Prediction (Acc) | Decision Making (Dec) | Treatment Effect Analysis (Trend) |
> | --- | --- | ---: | ---: | ---: |
> | Predict-only ensemble | Prediction without NL-to-task mapping | 0.7635 | 0.7151 | 0.6871 |
> | DeepSeek + TabPFN | LLM parsing + strong tabular predictor | 0.6278 | 0.6237 | 0.6857 |
> | DeepSeek main text-based | End-to-end free-form reasoning from table text | 0.5973 | 0.5699 | 0.5714 |
> | DeepSeek main with-tool | End-to-end reasoning with tool use on the paired table | 0.6021 | 0.5860 | 0.6762 |
>
> These comparisons make the scope of TOPBENCH more concrete. The **predict-only** baseline is an adaptive ensemble over strong tabular predictors, including **HistGradientBoosting, XGBoost, LightGBM, CatBoost, ExtraTrees, TabPFN**, which **does not solve the full benchmark**, because it is given the gold structured scenario and therefore does not need to infer the target, parse the user request, or build the correct prediction instance from language. It measures only the downstream tabular prediction difficulty. DeepSeek + TabPFN sits in between: the model must still map language into a structured predictive task, but the final predictor is replaced by a stronger tabular model.
>
> This comparison supports our main claim: TOPBENCH is not difficult only because tabular prediction is difficult. Performance rises substantially once the language-to-task burden is removed, and also improves when the downstream predictor is strengthened.
>
> We will integrate these baselines into the revised paper to make the benchmark boundary much clearer.
>
> ---
>
> >Q1: Please provide feature-count statistics.
>
> We have now added descriptive statistics for table structure. These additions will appear in the revision and appendix.
>
> **Overall benchmark statistics**
>
> | Statistic                             | Value                                                                                |
> | --- | --- |
> | Number of tables                      | 292                                                                                  |
> | `#rows` per table                     | min `160`, median `2,968`, mean `85,222.3`, max `6,778,266`                          |
> | `#columns` per table                  | min `5`, median `13`, mean `17.7`, max `55`                                          |
> | Column bucket share                   | `<=10`: `33.6%`, `11–20`: `26.7%`, `21–30`: `25.7%`, `31–40`: `11.0%`, `>40`: `3.1%` |
> | Tables with missing values            | `47.3%`                                                                              |
> | Avg missing-cell ratio over tables    | `2.85%`                                                                              |
> | Tables with categorical features      | `78.4%`                                                                              |
> | Avg numeric features per table        | `12.0`                                                                               |
> | Avg categorical features per table    | `5.84`                                                                               |
>
> These additions show that the benchmark spans a broad range of row scales, table widths, missing-value conditions, and schema types rather than concentrating on a narrow regime.

---

> > ### Author Rebuttal · Reviewer_s3Jk · 2026-04-03
> >
> > I thank the authors for their response, which has addressed my concerns. I now understand that TOPBENCH evaluates the language-to-prediction interface, which is somewhat distinct from traditional tabular prediction tasks in machine learning. Therefore, I believe the authors should shift the narrative focus toward this specific aspect rather than over-emphasizing the prediction task itself. Considering that the paper overlooks a significant body of prior work—which weakens the importance of the claimed contribution (Point 1, lines 118-120)—I have decided to maintain my current score.

---

> > > ### Author Response · Authors · 2026-04-07
> > >
> > > Thank you for the clarification and for carefully engaging with our rebuttal. We especially appreciate your explicit acknowledgement that the main concerns have been fully resolved.
> > >
> > > We would like to clarify one remaining point regarding Contribution 1 (Point 1, lines 118–120). Our intention there is not to claim novelty over tabular prediction as a whole. Rather, the paper defines a specific **implicit predictive TQA** setting within the TableQA space, where the answer is not explicitly present in the table and the model must go beyond retrieval to infer an unobserved outcome from historical data.
> > >
> > > This scope is stated consistently in the manuscript. In Figure 1 and the Introduction, the paper distinguishes traditional TableQA from cases where the answer is absent from the table and the model must infer it from historical patterns. In Section 3.1, we formalize this explicitly as a **two-stage inference problem**: first, the model performs intent abstraction from a natural-language query; second, it performs predictive inference over the paired historical table. The benchmark then instantiates this setting through four task families: Single-Point Prediction, Decision Making, Treatment Effect Analysis, and Ranking/Filtering. So the contribution is not “prediction” in the generic tabular ML sense, but a benchmarked **TQA problem formulation** for intent-grounded predictive reasoning over tables.
> > >
> > > We fully agree that prior work on tabular prediction is important and should be discussed more explicitly, and we already committed in the rebuttal to expand the related-work section and add strong traditional tabular baselines. Our point here is more focused: the manuscript is not positioning TOPBENCH against the entire tabular prediction literature, but against the gap between existing TQA benchmarks and this specific intent-grounded predictive setting. We hope this clarifies why we view Contribution 1 as a scoped TQA contribution rather than a broad claim over prediction in general.

---

### Official Review · Reviewer_y845 · 2026-03-11

**Soundness:** 2
**Presentation:** 3
**Significance:** 3
**Originality:** 2
**Overall Recommendation:** 4
**Confidence:** 4

**Summary:**

The paper introduces TOPBENCH, a benchmark for tabular question answering where the goal is not to retrieve facts already in the table, but to infer unobserved outcomes from historical data. It frames the problem as two steps. First the model has to infer the user’s intent and translate a natural query into a feature profile that is not literally present in the table. Then it has to perform predictive inference by learning a mapping from features to outcomes in the historical table. The benchmark contains 779 samples drawn from 35 tables across healthcare, finance, and daily consulting, and it covers four common predictive scenarios. These are single point prediction, decision making, treatment effect analysis, and ranking and filtering for batch style use cases. The evaluation mixes deterministic metrics for structured outputs with an LLM based judging pipeline for free form reasoning, including explicit verification steps meant to reduce judge hallucinations. Experiments compare direct text only reasoning against agentic workflows that generate code and use tools, and the analysis highlights recurring failure modes where models fall back to retrieval style heuristics instead of modeling, especially under long context pressure

**Compliance With Llm Reviewing Policy:**

Affirmed.

**Final Justification:**

Although my main concerns have mostly been addressed, I am keeping my original score because the paper would still benefit from another round of revision to reflect the rebuttal more clearly in the manuscript.

**Key Questions For Authors:**

Q1. What models are used for LLM as a Judge, and are the prompts and decoding settings fixed across all tasks and models.

Q2. Could there be model level bias when LLMs are used as judges, especially when the judge comes from the same family as some evaluated models.

Q3. How does reasoning get affected on implicit reasoning questions when fed a smaller tabular sample size, and does performance improve or degrade as the number of rows and columns changes.

Q4. How reliable is the judge pipeline in practice, and did the authors do any human spot checks or inter annotator style agreement checks on the judge outputs to validate scoring quality.

Q5. Since the benchmark pairs each query with the correct table, how would results change in a more realistic pipeline that includes a retrieval step to select the right table or subset of records before prediction.

**Limitations:**

Not completely, especially with a few metrics on their benchmark. They do discuss important technical scope limitations, for example that each query is paired with the relevant table and they do not evaluate retrieval or data discovery. They also highlight reliability issues like judge hallucination and add an automated verification pipeline.

**Constructive suggestions to improve the limitations**
- Add a short section on downstream harm for predictive tabular inference in high stakes domains like healthcare and finance, including risks of overconfident predictions, spurious correlations, and distribution shift.

- Clarify evaluation bias. Name the judge model and describe any human spot checks for judge outputs, since judge choice can systematically advantage certain model styles or verbosity.

**Strengths And Weaknesses:**

**Strength**

The evaluation design is thoughtful and tries to be robust to free form answers. The judge pipeline extracts structured predictions and also scores reasoning quality, and it adds a verification mechanism to limit judge hallucination. The benchmark spans three domains and is explicitly organized into four predictive task types that mirror common real-world scenarios. Good model coverage to analyze on all dimensions in this benchmark, including text only versus agentic workflows and analysis of tool use behaviors

**Weakness**

The paper does not clearly specify which judge model is used for the LLM as a judge pipeline, which makes it harder to reproduce the evaluation and to reason about judge bias. Human checking is described for validating generated samples during dataset construction, but it is not clearly described as a spot check for the judge outputs themselves, so it is unclear how often the judge evaluation was audited by people. The benchmark setup pairs each query with the relevant historical table, which is a reasonable intermediate step, but it leaves open how well the approach transfers to more realistic settings where systems must also discover or retrieve the right data before predicting

---

> ### Author Rebuttal · Authors · 2026-03-30
>
> Thank you for your thoughtful review and detailed comments. We provide our responses below.
>
> >W / Q1: The judge model is not clearly specified, and it is unclear whether prompts / decoding settings are fixed.
>
> We will make this explicit in the revision.
>
> | Judge setting | Value |
> | --- | --- |
> | Original main judge | deepseek-v3.2-instruct |
> | Alternate judge used for robustness check | gpt-5.2 |
> | Prompt scope | fixed by task and mode |
> | Hallucination-control design | Appendix Figure 7 |
>
> We will explicitly name the judge model in the main paper, release the exact prompt templates, and point more directly to Appendix Figure 7, which documents the clear extraction process.
>
> ---
>
> >Q2: Could there be model-family bias when LLMs are used as judges?
>
> To test the bias, we re-scored the same outputs with a second judge family (gpt-5.2) and compared the **extracted outcome**. This is the right comparison for judge faithfulness.
>
> **Task-level comparison**
>
> | Task / Setting | DeepSeek judge | GPT judge | Mean gap |
> | --- | ---: | ---: | ---: |
> | Single-Point Prediction, text-based | 0.5696 | 0.4934 | -0.0763 |
> | Single-Point Prediction, with-tool | 0.5901 | 0.5147 | -0.0754 |
> | Decision Making, text-based | 0.5699 | 0.5780 | +0.0081 |
> | Decision Making, with-tool | 0.5672 | 0.5618 | -0.0054 |
> | Treatment Effect Analysis, text-based | 0.5524 | 0.5905 | +0.0381 |
> | Treatment Effect Analysis, with-tool | 0.6095 | 0.6810 | +0.0714 |
> | **Overall** | **0.5763** | **0.5502** | **-0.0262** |
>
> The overall task-level gap is small (0.0262). As the logic score is designed for **fatal flaws**, we restrict the comparison to clearly low-logic cases (logic < 0.4), the two judges show 95.1% agreement on the low-logic flag overall. Therefore, we view this result as evidence that the judge-family effect on the core extracted outcome is limited.
>
> ---
>
> >Q3: How does reasoning change with smaller tabular samples, and with different numbers of rows and columns?
>
> We provide direct analyses along both the row and column dimensions.
>
> Across the full benchmark, the dataset distribution is:
>
> | Shape bucket | Share of all tables |
> | --- | ---: |
> | Short rows (≤1k) | 32.2% |
> | Medium rows (1k–100k) | 54.5% |
> | Long rows (>100k) | 13.4% |
> |
> | Narrow columns (≤10) | 33.6% |
> | Mid columns (11–30) | 52.4% |
> | Wide columns (>30) | 14.0% |
>
>
> **Results by row bucket**
>
> | Task / Setting | Short (≤1k) | Medium (1k–100k) | Long (>100k) |
> | --- | ---: | ---: | ---: |
> | Single-Point, text-based Acc | 0.5645 | 0.5009 | 0.5858 |
> | Single-Point, with-tool Acc | 0.5624 | 0.5091 | 0.5162 |
> | Decision Making, text-based Dec | 0.5910 | 0.5019 | 0.5377 |
> | Decision Making, with-tool Dec | 0.4801 | 0.4624 | 0.4927 |
> | Treatment Effect, text-based Trend | 0.4703 | 0.6221 | 0.5333 |
> | Treatment Effect, with-tool Trend | 0.4497 | 0.5907 | 0.5412 |
>
> **Results by column bucket**
>
> | Task / Setting | Narrow (≤10) | Mid (11–30) | Wide (>30) |
> | --- | ---: | ---: | ---: |
> | Single-Point, text-based Acc | 0.5255 | 0.5298 | 0.5990 |
> | Single-Point, with-tool Acc | 0.4922 | 0.5440 | 0.5146 |
> | Decision Making, text-based Dec | 0.4769 | 0.5253 | 0.6778 |
> | Decision Making, with-tool Dec | 0.3860 | 0.4766 | 0.6256 |
> | Treatment Effect, text-based Trend | 0.5396 | 0.5613 | — |
> | Treatment Effect, with-tool Trend | 0.4778 | 0.5884 | — |
>
> For our setting to generate more realistic queries, there is no wide-column dataset in Treatment Effect.
>
> Results show that the effect of table shape is real, but not monotonic. This suggests that the main challenge is not only context size, but also how well the model can organize under different data distributions, feature structures, and predictive task forms.
>
> Other statistics can be found at **Reviewer 4, Q1**.
>
> ---
>
> >Q4: How reliable is the judge pipeline in practice? Did you do human spot checks?
>
> We added a manual audit over 72 sampled cases from the three tasks.
>
> | Manual audit by task | Cases | Extraction agreement |
> | --- | ---: | ---: |
> | Single-Point Prediction | 24 | 100.0% |
> | Decision Making | 24 | 95.8% |
> | Treatment Effect Analysis | 24 | 100.0% |
> | **Overall** | **72** | **98.6%** |
>
> The extraction agreement is 98.6%; only one query have divergence because the response to be evaluated vaguely provided multiple options, while the judge strictly selected only one label. So both the human audit and the cross-judge comparison suggest that this extraction stage is reliable.
>
> ---
>
> >Q5: The benchmark pairs each query with the correct table, so how does it relate to a more realistic retrieval-first setting?
>
> We agree that a full deployment system would need retrieval before prediction. However, TOPBENCH is intentionally scoped as a controlled intermediate benchmark where the relevant historical table is already paired with the query. This isolates predictive reasoning from retrieval failure. We will make this scope boundary clearer earlier in the paper and position retrieval-first predictive inference as a next-step extension.

---

> > ### Author Rebuttal · Reviewer_y845 · 2026-04-03
> >
> > Although my main concerns have mostly been addressed, I am keeping my original score because the paper would still benefit from another round of revision to reflect the rebuttal more clearly in the manuscript.

---

> > > ### Author Response · Authors · 2026-04-07
> > >
> > > Thank you for this helpful follow-up. We appreciate that our rebuttal addressed most of your main concerns, and we also understand your view that the manuscript itself should more clearly reflect those clarifications. We will revise the paper accordingly in the final version by incorporating the rebuttal’s key clarifications. Thank you again for your constructive feedback and for the careful reading of our work.

---

### Official Review · Reviewer_b5fE · 2026-03-12

**Soundness:** 3
**Presentation:** 3
**Significance:** 4
**Originality:** 3
**Overall Recommendation:** 4
**Confidence:** 4

**Summary:**

This paper introduces TOPBENCH, a benchmark for evaluating LLMs on implicit predictive Table Question Answering. The benchmark comprises 779 samples across three domains and four tasks: Single-Point Prediction, Decision Making, Treatment Effect Analysis, and Ranking and Filtering. The authors evaluate 9 models under both text-based reasoning and agentic code-execution frameworks, using a hybrid evaluation pipeline that combines statistical metrics with an LLM-as-a-Judge approach.

**Compliance With Llm Reviewing Policy:**

Affirmed.

**Key Questions For Authors:**

- **Q1.** Could you provide performance results broken down by table scale (Short/Medium/Long as defined in Table 4)? Specifically, do models degrade uniformly as table size increases, or do certain tasks exhibit disproportionate sensitivity to scale?
- **Q2.** What exact truncation strategy was used for text-based reasoning on tables exceeding the context window? If head-truncation was applied, have you verified that the retained rows are representative of the full distribution? Would a different sampling strategy materially change the text-based results?
- **Q3.** Have you conducted any sensitivity analysis on the metric hyperparameters in Equations 1–2?
- **Q4.** Which specific model serves as the judge LLM for extraction and Logic Score evaluation? Can you release the exact prompt templates used for structured extraction and reasoning quality scoring?

**Limitations:**

yes

**Strengths And Weaknesses:**

## Strength

- **Well-motivated problem with practical relevance.** Real-world users routinely ask questions whose answers are not present in any table row, and existing benchmarks do not capture this. The paper formalizes this as a two-stage inference problem, which cleanly separates the contribution from prior work.
- **Task taxonomy is well-grounded and non-redundant.** The four sub-tasks each require a distinct reasoning capability.
- **Deliberate diversity in data characteristics.** The benchmark spans three domains, balances regression and classification objectives, and varies table scale from <1k to >6M rows.

## Weekness

- **No performance stratification by table scale.** The paper deliberately varies table size, but all results in Table 2 are aggregated across scales. Without per-scale breakdowns, it is impossible to determine whether failures stem from context truncation on large tables or from intrinsic modeling deficiency. The design choice to vary scale is wasted without this analysis.
- **Truncation strategy for text-based reasoning is unspecified.** Section 5.1 states large tables are truncated to fit context limits but never describes how rows are selected. Since text-based models can only reason over the rows they see, this undisclosed choice is a confound that could systematically bias results toward or against retrieval-based heuristics.
- **Metric hyperparameters are unjustified.** Equations 1–2 introduce $\lambda_1$=0.6, $\lambda_2$=0.4, $\tau$=2.0, and an unspecified decay rate $\alpha$, all without empirical justification or sensitivity analysis. No ablation demonstrates that reported rankings are robust to alternative settings.
- **Judge model identity and prompt templates are undisclosed.** The evaluation pipeline relies on an LLM-as-a-Judge for both structured extraction and Logic Score assignment. The specific model used is never named, and the scoring prompt is not provided. Since judge model choice and prompt wording are known to affect evaluation outcomes, reproducibility requires their disclosure.

---

> ### Author Rebuttal · Authors · 2026-03-30
>
> Thank you for your careful review and constructive feedback. We provide our responses below.
>
> >W1 / Q1: No performance stratification by table scale.
>
> The performance results broken down by table scale are as follow:
>
> **Average over per-model means**
>
> | Task / Setting | Short (≤1k) | Medium (1k–100k) | Long (>100k) |
> | --- | ---: | ---: | ---: |
> | Single-Point, text-based Acc | 0.5645 | 0.5009 | 0.5858 |
> | Single-Point, with-tool Acc | 0.5624 | 0.5091 | 0.5162 |
> | Decision Making, text-based Dec | 0.5910 | 0.5019 | 0.5377 |
> | Decision Making, with-tool Dec | 0.4801 | 0.4624 | 0.4927 |
> | Treatment Effect, text-based Trend | 0.4703 | 0.6221 | 0.5333 |
> | Treatment Effect, with-tool Trend | 0.4497 | 0.5907 | 0.5412 |
>
> **How many models are best in each bucket**
>
> | Task / Setting | Best on Short | Best on Medium | Best on Long |
> | --- | ---: | ---: | ---: |
> | Single-Point Prediction, text-based | 4 / 12 | 0 / 12 | 8 / 12 |
> | Single-Point Prediction, with-tool | 9 / 13 | 0 / 13 | 4 / 13 |
> | Decision Making, text-based | 8 / 12 | 1 / 12 | 3 / 12 |
> | Decision Making, with-tool | 5 / 13 | 2 / 13 | 6 / 13 |
> | Treatment Effect Analysis, text-based | 1 / 12 | 9 / 12 | 2 / 12 |
> | Treatment Effect Analysis, with-tool | 1 / 13 | 8 / 13 | 4 / 13 |
>
> These results show that table size matters, but its effect is **not uniform across tasks**. The pattern changes with **task type, data distributions, and system mode**. Combined with the baseline results in **Reviewer 1, Q2**, we believe the more central difficulty is not only long-context handling, but the broader challenge of mapping a natural-language predictive request into the correct prediction problem and modeling properly to solve the various prediction problems.
>
> ---
>
> >W2 / Q2: The truncation strategy is unspecified.
>
> We compared three row-retention strategies, always keeping the table header:
>
> - head
> - random (**the strategy used in the paper**)
> - stratified (uniformly sample rows from each chunk of the table to preserve positional coverage)
>
> We average over the two models that we used throughout this truncation analysis (DeepSeek-V3.2-Ins and Qwen3-Ins). The truncation ablation was run **only on the subset of tables that actually required truncation**.
>
> **Truncation ablation**
>
> | Task | Head | Random | Stratified |
> | --- | ---: | ---: | ---: |
> | Single-Point Prediction Acc | 0.5556 | 0.6064 | 0.6135 |
> | Decision Making Dec | 0.5476 | 0.5476 | 0.5238 |
> | Treatment Effect Analysis Trend | 0.5500 | 0.5667 | 0.5333 |
>
> Different row-retention choices change, which evidence is visible to the model, so they can change the final answer. However, due to opportunities for long-context hallucinated or weakly grounded reasoning, this impact is ultimately limited.
>
> ---
>
> >W3 / Q3: Metric hyperparameters are unjustified.
>
> We agree that the configuration should be made explicit. The sensitivity test will be added in Section 4.
>
> **Full sensitivity grid**
> - point weight ∈ **{0.2, 0.3, 0.4, 0.5, 0.6, 0.7, 0.8}**
> - interval weight = **1 − point weight**
> - width factor ∈ **{1.5, 2.0, 3.0}**
> - width decay rate ∈ **{0.5, 1.0, 2.0}**
>
> We ran this full stress test on **Single-Point Prediction**, where the interval-scoring component enters most directly.
>
> | Single-Point Prediction sensitivity result | text-based | with-tool |
> | --- | ---: | ---: |
> | Top-1 unchanged across all profiles | 1.0 | 1.0 |
> | Rank agreement vs. default, mean | 0.9960 | 0.9913 |
> | Pairwise order agreement, mean | 0.9900 | 0.9833 |
>
> Rank agreement vs. default means the average Spearman rank correlation across all tested profiles.
> Pairwise order agreement means the average fraction of model pairs whose relative order remains unchanged compared with the default profile.
>
> These hyperparameters do affect the **absolute score scale**, but the **relative ranking is extremely stable**.
>
> ---
>
> >W4 / Q4: The judge model identity and prompt templates are undisclosed.
>
> This was mainly omitted from the main text due to space, and we will make it explicit in Section 3.
> - The original main judge is **deepseek-v3.2-instruct**
> - The prompt templates are **fixed by task and mode**
> - We will release the exact prompts used
>
> We will also point more directly to **Appendix Figure 7**, which describes the clear extraction process.
>
> We added two direct checks:
>
> | Judge reliability / robustness check | Result |
> | --- | ---: |
> | Manual Extraction agreement | 98.6% |
> | Mean task-score gap between **gpt-5.2 and deepseek-v3.2-ins** judge | 0.0262 |
>
> The extraction agreement is 98.6%; **only one query** have divergence because the response to be evaluated vaguely provided multiple options, while the judge model strictly selected only one classification label. We therefore believe the extraction stage is highly reliable. The remaining judge-to-judge difference is mainly in the **logic** term, which is only used to penalize clearly incoherent bad cases, not to drive the main ranking among otherwise valid answers.

---

> > ### Author Rebuttal · Reviewer_b5fE · 2026-04-04
> >
> > Thank you for your response. Several concerns have been addressed, but two points require further clarification.
> >
> > Q1. The reported numbers are averaged across all models, which collapses inter-model variance entirely. My original question asked whether specific models fail disproportionately at particular scales. A model × scale cross-tabulation is needed to answer this; aggregate means cannot.
> >
> > Q2. The truncation ablation was run on only 2 of 9 evaluated models, which is insufficient for generalization. Additionally, the subset of tables that actually required truncation is not quantified. What fraction of the 779 samples falls into this subset? Without this, the scope of the ablation's applicability to the main results remains unclear.

---

> > > ### Author Response · Authors · 2026-04-07
> > >
> > > Thank you for your careful review and constructive feedback. We appreciate this follow-up, and we provide a more complete response below.
> > >
> > > > Q1:The reported numbers are averaged across all models, which collapses inter-model variance entirely.
> > >
> > > In our earlier rebuttal, due to space limits, we only reported bucket-level summaries. We now provide the **model×scale** cross-tabulation to expose inter-model variance directly. The row buckets are **Short(<=1k), Medium(1k-100k), and Long(>100k)**.
> > >
> > > We also clarify that the earlier 12/13 counts came from a broader export that included with-info prompt variants. Here, to keep the comparison focused and within the rebuttal length limit, we report only the core text-based models. We do not repeat the with-tool table here because the scale question is less directly tied to the code setting.
> > >
> > > **Text-based models**
> > >
> > > |Model|Single-Point(S/M/L)|Decision Making(S/M/L)|Treatment Effect(S/M/L)|
> > > |---|---|---|---|
> > > |DeepSeek-V3.2-Ins|0.63/0.53/0.65|0.57/0.58/0.55|0.47/0.70/0.57|
> > > |DeepSeek-V3.2-Think|0.61/0.50/0.68|0.67/0.59/0.57|0.50/0.72/0.50|
> > > |Gemini3.0-Flash|0.73/0.59/0.71|0.74/0.57/0.60|0.56/0.70/0.57|
> > > |GPT-5.2|0.54/0.54/0.58|0.63/0.52/0.60|0.41/0.56/0.57|
> > > |Qwen3-Ins|0.62/0.54/0.57|0.59/0.51/0.50|0.47/0.67/0.47|
> > > |Qwen3-Think|0.52/0.40/0.43|0.43/0.37/0.41|0.47/0.47/0.43|
> > > |TableLLM-8B|0.36/0.36/0.43|0.57/0.38/0.48|0.47/0.61/0.47|
> > > |TableLLM-13B|0.22/0.27/0.36|0.39/0.37/0.43|0.34/0.30/0.43|
> > >
> > > These cross-tabs show that the effect of scale is clearly task- and model-dependent. For example, in Single-Point, some peak on long(DeepSeek-Ins,DeepSeek-Think,GPT-5.2), while others peak on short(Gemini,Qwen-Ins,Qwen-Think). In Treatment Effect, most models peak on medium, while GPT-5.2 peaks on long. So the main difficulty cannot be explained by context length alone; it also depends on model family, task form, and data distribution.
> > >
> > > ---
> > >
> > > > Q2:The truncation ablation was run on only 2 of 9 evaluated models, and the subset of affected tables is not quantified.
> > >
> > > We agree that the scope should be stated.
> > >
> > > |Scope|Truncated|Total|Fraction|
> > > |---|---:|---:|---:|
> > > |All benchmark queries|136|779|17.5%|
> > > |Predictive tasks only|136|565|24.1%|
> > >
> > > |Predictive task|Truncated|Total|Fraction|
> > > |---|---:|---:|---:|
> > > |Single-Point|64|274|23.40%|
> > > |Decision Making|42|186|22.60%|
> > > |Treatment Effect|30|105|28.60%|
> > >
> > > So the truncation analysis applies to a specific subset of the benchmark.
> > >
> > > We also clarify the model scope. With-tool systems are not included because truncation of serialized table context is not the relevant bottleneck for them; they write code and read the table directly from file.
> > >
> > > **Scores on the truncation-affected subset only**
> > >
> > > |Model|Single-Point: Head/Random/Stratified|Decision Making: Head/Random/Stratified|Treatment Effect: Head/Random/Stratified|
> > > |---|---|---|---|
> > > |DeepSeek-V3.2-Ins|0.56/0.63/0.65|0.55/0.57/0.57|0.60/0.60/0.53|
> > > |DeepSeek-V3.2-Think|0.59/0.57/0.56|0.60/0.48/0.62|0.40/0.67/0.67|
> > > |Qwen3-Ins|0.55/0.58/0.58|0.55/0.52/0.48|0.50/0.53/0.53|
> > > |GPT-5.2|0.58/0.53/0.59|0.64/0.60/0.60|0.50/0.50/0.60|
> > >
> > > To connect this subset back to the main benchmark, we also report the full text-based benchmark under head, random, and stratified.
> > >
> > > **Scores on the full text-based benchmark under different truncation strategies**
> > >
> > > |Model|Single-Point: Head/Random/Stratified|Decision Making: Head/Random/Stratified|Treatment Effect: Head/Random/Stratified|
> > > |---|---|---|---|
> > > |DeepSeek-V3.2-Ins|0.57/0.58/0.59|0.56/0.57/0.57|0.59/0.59/0.57|
> > > |DeepSeek-V3.2-Think|0.56/0.53/0.54|0.56/0.51/0.56|0.54/0.62/0.62|
> > > |Qwen3-Ins|0.56/0.57/0.56|0.54/0.53/0.52|0.54/0.55/0.55|
> > > |GPT-5.2|0.53/0.53/0.55|0.56/0.55/0.56|0.50/0.50/0.57|
> > >
> > > We need to clarify why some other models are not included in this focused truncation table:
> > > - **Claude** is not included because we only tested its code capability in our paper.
> > > - **Gemini** is not included because the results remained incomplete under API-side instability, so we prefer not to report partial evidence.
> > > - **Qwen3-Think** is not included because, **as noted in our paper**, long text-based prompting frequently led to empty final answers, likely because the model entered a **repetitive long-chain reasoning pattern** that consumed the response budget before a final answer was produced. **DeepSeek-Think** shows this issue much less often, so we keep it as an additional robustness check.
> > > - **TableLLMs** are not included in this focused truncation discussion because their scores are much lower and they are less informative for comparing truncation sensitivity among the stronger text-based systems.
> > >
> > > Therefore, truncation strategy matters only for the subset that actually exceeds the context budget; on that subset, changing row-retention policy changes which evidence is visible and can therefore change the final answer. But when we return to the full text-based benchmark, the effect remains bounded and task-dependent, rather than materially changing the overall conclusions.

---

### Official Review · Reviewer_5Jgt · 2026-03-13

**Soundness:** 3
**Presentation:** 3
**Significance:** 3
**Originality:** 3
**Overall Recommendation:** 4
**Confidence:** 3

**Summary:**

This paper introduces TOPBENCH, a new benchmark for implicit predictive tabular question answering. Unlike conventional Table QA, which mainly focuses on retrieving or aggregating facts explicitly stored in tables, this work studies a more realistic setting where the answer is not directly contained in the table. Instead, the model must first recognize the user’s latent predictive intent and then reason over historical tabular data to perform prediction, decision making, treatment-effect analysis, or ranking/filtering. To support this, the authors build a benchmark with 779 examples from 35 source tables across domains such as healthcare, finance, and daily consulting, and design an evaluation framework for both natural-language and structured outputs. Experiments show that the main challenge for current models is not only limited predictive accuracy, but also the failure to recognize that these queries require prediction rather than lookup. Overall, the paper moves Table QA beyond explicit fact extraction toward implicit predictive reasoning over tables, and provides a benchmark, task taxonomy, and evaluation pipeline for this setting.

**Compliance With Llm Reviewing Policy:**

Affirmed.

**Key Questions For Authors:**

1. The paper states that the data was constructed through a multi-stage synthetic process, combined with LLM review and human verification. However, it remains unclear to what extent these samples approximate predictive tabular questions posed by real users. Could the authors provide more direct evidence, such as a small number of real query controls, human authenticity assessments, or comparisons with real task distributions?

2.Current experiments primarily compare different LLM and agent workflows, but it remains unclear whether TOPBENCH's challenges stem primarily from “mapping natural language problems to prediction tasks” or from “the inherent difficulty of prediction itself”. Could the authors provide stronger traditional prediction baselines?

3.The paper concludes that the current default configuration pairs relevant historical tables with queries, thus remaining a controlled intermediate step. How do the authors view the relationship between TOPBENCH and a more complete, real-world workflow? For example, are there future plans to expand to a configuration where “relevant tables are retrieved first, followed by predictive inference”?

**Limitations:**

Yes.

**Strengths And Weaknesses:**

Strengths
S1.The problem setting holds practical significance. This paper focuses not on the common table-based retrieval or aggregation tasks in traditional Table QA, but rather on scenarios where “the answer does not directly exist in the table and requires predictive reasoning based on historical data”. This setting more closely aligns with real-world user needs when interacting with tables, making the topic itself highly valuable in practical terms. It also addresses the shortcomings of existing Table QA benchmarks, which primarily focus on explicit fact-based question answering.
S2. The paper is generally well written and easy to follow.
S3. This paper addresses a critical and practical issue. Many real-world table interactions are not purely retrieval tasks but rather predictive or decision-oriented queries expressed in natural language. In this sense, it reveals a genuine gap between standard table question-answering benchmarks and the types of tasks users actually expect systems to perform.

Weaknesses
W1. Although the baseline data underwent post-validation, it was still synthesized from curated tables via prompts. Consequently, the authenticity of language distribution and task frequency remains partially dependent on the generation pipeline rather than being entirely based on natural user logs. While some core conclusions  are supported by experiments, they primarily rest on empirical correlations rather than complete causal relationships.

W2. The primary shortcoming of this paper lies in certain methodological sections that appear overly complex, exceeding the scope of thorough justification within the main text. Specifically, the design choices, such as employing large language models as evaluators, interval-based regression scoring, and hallucination verification mechanisms, are quite sophisticated. However, the paper should more clearly articulate the degree to which its conclusions depend on these design decisions. Additionally, the opening phrase “Decision Making” is missing from the third paragraph of Section 3.1.

W3.This paper assumes that relevant historical tables are already provided, thus primarily evaluating whether implicit predictive reasoning can be performed after obtaining the tables. It does not assess the more comprehensive real-world workflow, such as first identifying suitable data sources, then understanding user intent, and finally completing analysis and prediction.

---

> ### Author Rebuttal · Authors · 2026-03-30
>
> Thank you for your encouraging review and thoughtful questions. We provide our responses below.
>
> > W1 / Q1: Synthetic data may not reflect real user predictive queries.
>
> We agree that TOPBENCH is not a raw user-log benchmark, and we will state this more clearly. At the same time, it is not built from unconstrained synthetic prompts. As described in **Section 3**, we reviewed and collected real discussions from public forums where users sought prediction-like help from past records or comparisons. These sources include **Reddit, MoneySavingExpert, Edmunds, Mayo Clinic Connect, Cancer Chat, etc**. Then we abstracted predictive intents and organized them into the four tasks: Single-Point Prediction, Decision Making, Treatment Effect Analysis, and Ranking and Filtering. All queries were manually verified and corrected after generation.
>
> To support this, we added a human check. It was a5-annotator audit over 48 sampled queries. Each query was rated on three criteria:
>
> - **Authenticity**: whether it reflects a real predictive need
> - **Naturalness**: whether it looks like a real user query
> - **Plausibility**: whether it is reasonable for the paired table and task setup
>
> | Dimension | Value |
> | --- | ---: |
> | Authenticity | 4.00 / 5 |
> | Naturalness | 4.00 / 5 |
> | Plausibility | 4.75 / 5 |
>
> These results support our main point: the benchmark is **derived from real predictive user intents** and remains close to realistic predictive table interaction. In the revision, we will also include **representative real-user queries** in the appendix, so that readers can more directly inspect this grounding process.
>
> ---
>
> >W2: The evaluation design is complex, so it is unclear how much the conclusions depend on it.
>
> We will explain this more clearly. We will separate three roles in evaluation:
>
> 1. In Single-Point, Decision Making, and Treatment Effect, the LLM is used only to extract the predicted value or category from free-form text, while the **final accuracy metric** is computed by **deterministic rules**. In Ranking and Filtering, no LLM is used in evaluation; we directly use specific metrics to measure ranking, retrieval, and filtering quality.
> 2. The **hallucination-control mechanism** in **Appendix Figure 7** improves faithfulness when extracting.
> 3. The **logic score** is only a safeguard for clearly bad cases, mainly catching self-contradictory, vacuous, or incoherent outputs.
>
> We conducted a **manual extraction audit** to verify whether the extracted structured values matched the model outputs and we also evaluated the same cases with **DeepSeek-V3.2-Ins** and **GPT-5.2** to test sensitivity to extraction, which prove the results are reliable; more details are provided in **Reviewer 3, Q2 and Q4**.
>
> We will also fix the missing “Decision Making” phrase in Section 3.1.
>
> ---
>
> >Q2: Are the benchmark difficulties mainly about NL-to-task mapping, or about prediction itself? Add stronger prediction baselines.
>
> We agree this distinction matters. To address it, we will add baselines that isolate different parts of the pipeline in Section 6.
>
> | Method | Pipeline | Single-Point (Acc) | Decision Making (Dec) | Treatment Effect (Trend) |
> | --- | --- | ---: | ---: | ---: |
> | DeepSeek text-based | LLM reads text and answers directly | 0.5973 | 0.5699 | 0.5714 |
> | DeepSeek with tool | LLM + tool use | 0.6021 | 0.5860 | 0.6762 |
> | Predict-only | Uses the gold structured data directly; no NL parsing | 0.7635 | 0.7151 | 0.6871 |
> | DeepSeek + TabPFN | DeepSeek parses the query into a structured payload, then TabPFN predicts | 0.6278 | 0.6237 | 0.6857 |
>
> The **predict-only** baseline is an adaptive ensemble over strong tabular predictors, including **HistGradientBoosting, XGBoost, LightGBM, CatBoost, ExtraTrees, TabPFN**. It receives the gold structured data directly without the NL parsing. Its gains show that a major part of the challenge comes from mapping a free-form request into the right structured task.
>
> It is still not perfect because the tables come from real and difficult Kaggle datasets, many with **long-tail distributions, class imbalance, noise, mixed feature types, etc.** This is exactly our motivation: TOPBENCH combines two real difficulties:
> - tabular prediction itself,
> - mapping nl predictive intent into the right prediction task.
>
> ---
>
> >W3 / Q3: The relevant table is already given, so the benchmark does not cover the full real workflow.
>
> We agree. As stated in the paper, TOPBENCH is a controlled intermediate benchmark: the relevant historical table is explicitly paired with the query so that we can isolate predictive reasoning after the right data is identified. We chose this to avoid mixing two distinct failures: retrieving the wrong table or failing to reason predictively over the correct one. Even in this setting, current LLMs still often fall back to retrieval-style heuristics. We will make this scope boundary clearer earlier in the paper and present retrieval-first predictive inference as a next-step extension.

---

> > ### Author Rebuttal · Reviewer_5Jgt · 2026-04-04
> >
> > Thank you for the thoughtful response. I look forward to seeing the corresponding revisions incorporated into the final version of the paper. While I acknowledge the contributions of this work, certain aspects require further clarification and refinement. Therefore, I will maintain my current rating of Weak Accept.
> > Thanks！

---

> > > ### Author Response · Authors · 2026-04-07
> > >
> > > Thank you for your thoughtful follow-up and for maintaining your support for the paper. We appreciate your recognition of the contribution, as well as your careful comments on the aspects that still need clarification and refinement. We will revise the final version accordingly by incorporating the clarifications, additional analyses, and positioning changes discussed in the rebuttal, so that the paper more clearly communicates its scope, methodology, and contribution. Thank you again for your constructive feedback and for your time.

---

### Decision · Program_Chairs · 2026-04-30

**Decision:**

Accept (regular)

**Comment:**

After a strong rebuttal from the authors, there is consensus among reviewers that this paper should be accepted. There is strong agreement that this is an important research direction and the authors provide a good solution for addressing the direction.

The authors should make improvements to the manuscript as promised in the rebuttal phase, including the addition of a related work section on tabular prediction.